# Endocytic trafficking determines cellular tolerance of presynaptic opioid signaling

Damien Jullié[1,2]*, Camila Benitez[3], Tracy A Knight[3], Milos S Simic[1], Mark von Zastrow[1,2,4]*

[1]Department of Cellular and Molecular Pharmacology, University of California, San Francisco School of Medicine, San Francisco, United States; [2]Department of Psychiatry and Behavioral Sciences, University of California, San Francisco School of Medicine, San Francisco, United States; [3]Tetrad graduate program, University of California, San Francisco, San Francisco, United States; [4]Quantitative Biology Institute, University of California, San Francisco, San Francisco, United States

**Abstract** Opioid tolerance is well-described physiologically but its mechanistic basis remains incompletely understood. An important site of opioid action in vivo is the presynaptic terminal, where opioids inhibit transmitter release. This response characteristically resists desensitization over minutes yet becomes gradually tolerant over hours, and how this is possible remains unknown. Here, we delineate a cellular mechanism underlying this longer-term form of opioid tolerance in cultured rat medium spiny neurons. Our results support a model in which presynaptic tolerance is mediated by a gradual depletion of cognate receptors from the axon surface through iterative rounds of receptor endocytosis and recycling. For the μ-opioid receptor (MOR), we show that the agonist-induced endocytic process which initiates iterative receptor cycling requires GRK2/3-mediated phosphorylation of the receptor's cytoplasmic tail, and that partial or biased agonist drugs with reduced ability to drive phosphorylation-dependent endocytosis in terminals produce correspondingly less presynaptic tolerance. We then show that the δ-opioid receptor (DOR) conforms to the same general paradigm except that presynaptic endocytosis of DOR, in contrast to MOR, does not require phosphorylation of the receptor's cytoplasmic tail. Further, we show that DOR recycles less efficiently than MOR in axons and, consistent with this, that DOR tolerance develops more strongly. Together, these results delineate a cellular basis for the development of presynaptic tolerance to opioids and describe a methodology useful for investigating presynaptic neuromodulation more broadly.

*For correspondence:
damien.jullie@gmail.com (DJ);
mark@vzlab.org (MZ)

Competing interest: The authors declare that no competing interests exist.

## Editor's evaluation

This manuscript examines the inhibition of transmitter release induced by the activation of opioid receptors, both MOR and DOR, using a novel imaging method. The authors specifically examine how the inhibition of transmitter release is changed following prolonged exposure to saturating concentrations of agonists and they showed convincingly that there is a depletion of plasma membrane-associated receptors and suggest that the decline in receptors at the plasma membrane underlies presynaptic tolerance. This work addresses a long-standing question about how tolerance develops at the presynaptic level and indicates that the location of receptors is critically important in the development of tolerance. This work is fundamental and a game changer in the understanding of tolerance at the cellular level.

## Introduction

The development of physiological tolerance to opioid agonists provides a fascinating example of neurobehavioral plasticity initiated through the activation of specific G protein-coupled receptors (GPCRs). It is also clinically significant because tolerance limits the therapeutic utility of opioid drugs. While opioid agonists are highly effective in the acute management of pain, maintaining analgesic efficacy under conditions of prolonged or repeated administration tends to require escalating doses. Tolerance develops to other physiological effects of opioids as well, such as the suppression of central ventilatory drive underlying the clinical phenomenon of opioid-induced respiratory depression (OIRD), but typically over a longer period of time (*Athanasos et al., 2006*; *Hayhurst and Durieux, 2016*; *Paronis and Woods, 1997*). This kinetic 'mismatch' in the development of tolerance to various opioid-induced effects narrows the therapeutic window for analgesia and is arguably a root cause of the present epidemic of opioid drug-related deaths. Therefore, an important goal of fundamental research is to more fully understand how opioids produce physiological adaptations which develop at widely different rates.

Part of the answer undoubtedly lies in the complexity of in vivo opioid physiology. Opioids are well known to impact neural function at multiple levels, from molecular mechanisms that occur in discrete receptor-expressing neurons to adaptations which propagate through synaptic networks and neural circuits (*Cahill et al., 2016*; *Corder et al., 2018*). Even for mechanisms resolved in individual cells, however, it has long been recognized that adaptations can develop at different rates (*Chavkin and Goldstein, 1982*; *Law et al., 1982*; *Sharma et al., 1975*). Accordingly, one plausible approach toward elucidating kinetic differences among opioid-induced neuroadaptations is to focus on mechanisms that occur in individual neurons but produce physiological effects spanning a range of timescales.

Agonist-induced phosphorylation of receptors is one such mechanism. In particular, phosphorylation of the µ-type opioid receptor (MOP-R or MOR) cytoplasmic tail by GPCR kinases (GRKs) mediates desensitization of MOR-mediated control of potassium channels, a response determining the postsynaptic excitability of neurons (*Arttamangkul et al., 2018*; *Williams et al., 2013*). This desensitization process characteristically develops over minutes (*Blanchet and Lüscher, 2002*; *Harris and Williams, 1991*; *Lowe and Bailey, 2015*; *Williams et al., 2013*), consistent with the time course of MOR phosphorylation and subsequent phosphorylation-dependent endocytosis of MOR in neurons (*Arttamangkul et al., 2008*; *Just et al., 2013*). However, phosphorylation of the MOR tail, and on the same Ser/Thr residues required for rapid desensitization, has been clearly shown to attenuate physiological opioid actions after chronic as well as acute administration (*Kliewer et al., 2019*). Might there be an additional cellular locus at which phosphorylation of the MOR tail drives the development of opioid tolerance over a longer time period?

A possible locus is the presynaptic terminal, where a key physiological action of opioids is to inhibit vesicular neurotransmitter release. Presynaptic inhibition is characteristically resistant to desensitization when assessed over minutes (*Blanchet and Lüscher, 2002*; *Fyfe et al., 2010*; *Jullié et al., 2020*; *Lowe and Bailey, 2015*; *Pennock et al., 2012*; *Rhim et al., 1993*) but has been shown to develop tolerance after prolonged opioid exposure (*Fyfe et al., 2010*; *Lowe and Bailey, 2015*). Nevertheless, MOR was recently shown to undergo phosphorylation-dependent endocytosis in presynaptic terminals within minutes (*Jullié et al., 2020*). Together, these observations suggest the possibility that phosphorylation of the MOR cytoplasmic tail, despite not producing a rapid desensitization of opioid signaling at the presynapse, drives the development of this slower form of presynaptic opioid tolerance.

Here, we describe an experimental approach to explicitly test this hypothesis. We delineate a primary culture system enabling the direct measurement of presynaptic tolerance and show that phosphorylation of the MOR cytoplasmic tail is indeed required for this adaptation. We propose a simple cellular mechanism, based on iterative receptor recycling, that is sufficient to explain how rapid phosphorylation of MOR produces presynaptic tolerance over an extended time scale. We then show that a similar model applies to the development of tolerance to presynaptic inhibition by the homologous δ-type opioid receptor (DOP-R or DOR) except that, remarkably, DOR endocytosis in axons does not require phosphorylation of the receptor cytoplasmic tail. Our results provide fundamental insight into the question of how opioid-induced neuroadaptations develop over distinct timescales and contribute a methodology that we anticipate will facilitate the study of presynaptic neuromodulation more generally.

## Results

### Presynaptic tolerance to opioids is associated with a loss of surface opioid receptors in the axon

We assayed opioid-induced presynaptic inhibition by adapting a widely used pHluorin-based unquenching assay (*Sankaranarayanan et al., 2000*) to monitor opioid effects on presynaptic activity in cultured neurons. In this assay, neurons were expressing opioid receptors together with VAMP2-SEP, imaged using a widefield microscope, and were electrically stimulated to induce synaptic vesicle exocytosis. The super-ecliptic pHluorin (SEP) is a pH sensitive GFP whose fluorescence increases as the synaptic vesicle protein VAMP2-SEP relocalizes from acidic synaptic vesicles to the terminal plasma membrane. This fluorescence increase provides a readout for presynaptic activity, which is typically lower when neurons are perfused with agonist for opioid receptors, reflecting opioid mediated presynaptic inhibition. The basic hardware configuration is summarized in *Figure 1A*. Details of a lab-built apparatus and an automated data analysis pipeline, including specific code modules, are included in Appendix 1. In the adult striatum, a large fraction of medium spiny neurons express MOR or DOR endogenously. However, in our primary neuron cultures, only a small proportion of neurons express opioid receptors endogenously, as assessed functionally and by immunocytochemistry (*Jullié et al., 2020*). Therefore, co-expression of recombinant receptors together with the synaptic vesicle exocytosis reporter is necessary to detect presynaptic inhibition using the aggregate readout. *Figure 1B* shows an example of a recording from the analysis and illustrates how the degree of presynaptic inhibition was defined. We believe this simple assay offers a number of advantages for mechanistic interrogation, relative to more complex models that offer advantages for relating presynaptic inhibition to physiology. First, optical measurement of presynaptic activity provides a direct and reliable readout of the degree of inhibition that is independent of compounded postsynaptic effects. Second, the cultured neuron system is highly amenable to genetic and pharmacological manipulations. Third, the hardware and analysis pipeline are simple and largely open-source, facilitating rapid and economical deployment.

To examine the effect of prolonged opioid exposure in this system, we measured the presynaptic inhibition mediated by [D-Ala$^2$, $N$-MePhe$^4$, Gly-ol]-enkephalin (DAMGO), a peptide full agonist of MOR. We compared inhibition of the electrically-evoked pHluorin response observed in opioid-naïve neurons (we define this as the acute condition) to that observed in neurons pre-exposed to DAMGO for 18 hr (we define this as the chronic condition). Significant inhibition was observed in both conditions (unpaired t-test compared to control, acute $p<1 e^{-5}$, chronic $p<1 e^{-5}$), but the degree of inhibition was reduced in the chronic condition (*Figure 1C*, unpaired t-test $p<1 e^{-5}$). These results indicate that prolonged agonist exposure promotes tolerance to presynaptic inhibition by opioids. We further assessed this by concentration-response analysis, verifying decreased efficacy of presynaptic inhibition and also revealing a decrease in potency (*Figure 1D*, EC50 acute 1.13 nM, EC50 after induction of tolerance 33.85 nM).

Presynaptic inhibition by opioids is well known to be resistant to rapid desensitization processes which attenuate signaling typically over several minutes (*Blanchet and Lüscher, 2002*; *Fyfe et al., 2010*; *Jullié et al., 2020*; *Lowe and Bailey, 2015*; *Pennock et al., 2012*; *Rhim et al., 1993*), suggesting that presynaptic tolerance represents a distinct regulatory process. In addition, after the induction of opioid tolerance in vivo, presynaptic inhibition remains resistant to rapid desensitization while desensitization of the postsynaptic response is enhanced (*Arttamangkul et al., 2018*; *Fyfe et al., 2010*). In our in vitro system, we did not detect any evidence for desensitization of the DAMGO response over a 10-min interval of sequential stimulation after the induction of tolerance (*Figure 1E*, left. Mean inhibition at 2 min 36.98 ± 2.80%, mean inhibition at 12 min 37.44 ± 3.34%, paired t-test, p=0.87). Rather, time course analysis revealed that tolerance develops gradually over multiple hours (*Figure 1F*). This extended time course is reminiscent of the process of receptor downregulation, described previously in other systems and associated with a depletion of the total receptor reserve (*Chavkin and Goldstein, 1982*; *Christie, 2008*; *Law et al., 1984*). Supporting the hypothesis that presynaptic tolerance involves a similar process, we found that reducing receptor reserve using the irreversible antagonist β-Chlornaltrexamine (β-CNA) accelerated the development of presynaptic tolerance (*Figure 1F*, unpaired t-test, p=0.075, 0.002, 0.047 for acute, 2 and 4 hr, respectively). This effect was quite sensitive, with significant acceleration evident even under alkylation conditions that have only a small impact on the maximal opioid response and which produce no detectable rapid desensitization of

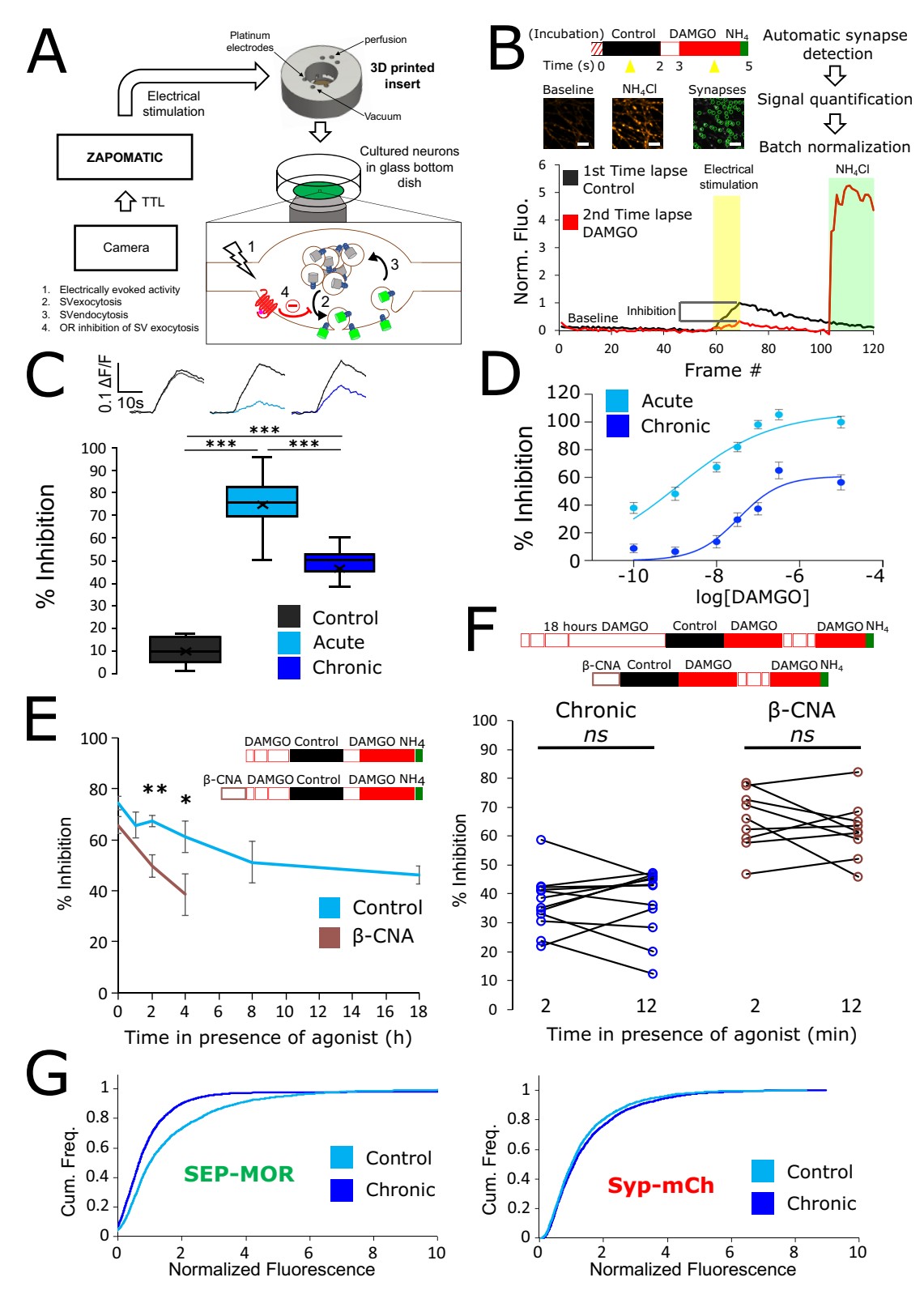

**Figure 1.** Loss of MOR mediated presynaptic inhibition under chronic activation conditions is paralleled by a reduction in surface receptor number in axons. (**A**) Schematic of the experimental setup, with highlighted open-source hardware used for electrical stimulation in synchronicity with image acquisition (zapomatic) and perfusion of solution onto cultured primary cultured neurons (3D printed insert) transfected with opioid receptors and VAMP2-SEP. The enlarged diagram depicts the biological process of electrically stimulated synaptic vesicle recycling (1) monitored with widefield

*Figure 1 continued on next page*

*Figure 1 continued*

fluorescence microscopy. Exocytosis of VAMP2-SEP containing synaptic vesicles causes (2) an increase in fluorescence intensity (green), which returns to baseline after recapture of VAMP2-SEP by endocytosis (3) and quenching of the fluorescence (gray). Active opioid receptors (red) inhibit exocytosis of synaptic vesicles (4). (**B**) Description of the experiment design and automated analysis pipeline. For measurement of acute inhibition, neurons are directly placed in imaging solution on the imaging system. For measurement of inhibition after chronic treatment, neurons are pre-treated with agonist for 18 hr (unless specified otherwise). A first time lapse (120 frames, 1 Hz) is acquired in control imaging solution (black box, black curve) and neurons are electrically stimulated at 10 Hz for 10 s 1 min into the time lapse. One minute after perfusion of a solution containing DAMGO 10 µM (open red box), a second time lapse is acquired (red box, red curve) with the same electrical stimulation and imaging paradigm, and for the last 20 frames the solution is exchanged for ammonium chloride (NH$_4$Cl). A differential image NH$_4$Cl – baseline is used to automatically detect putative synapses, representative images are shown. Signal is quantified over multiple tens of putative synapses and used to validate real synapses. Normalized data are pooled for the same condition. For each acquisition, we obtain curves as depicted after normalization by the maximum amplitude of the control condition (n=508 synapses for this acquisition). Note the difference in maximal amplitude in the presence of DAMGO compared to control, which reflects inhibition of synaptic vesicle exocytosis by opioid receptors. Scale bar is 10 µm. (**C**) Upper panel shows average fluorescence curves normalized over NH$_4$Cl (ΔF/F) for all synapses, lower panel displays percentage whisker plots of inhibition of SV exocytosis for each acquisition (4 quartiles +mean marker "X"). Inhibition of SV exocytosis, compared to control baseline as explained in B, for cells perfused with control solution (Control, inset n=1,603 synapses, n=6 acquisitions), cells perfused with DAMGO 10 µM (Acute, inset n=3,236 synapses, n=20 acquisitions), and cells pretreated with DAMGO 10 µM for 18 hr and perfused with DAMGO (Chronic, inset n=2025 synapses, n=13 acquisitions). (**D**) Normalized concentration-response curves of MOR mediated presynaptic inhibition acutely or after the induction of tolerance (Acute n=6/6/10/9/10/9, Chronic n=19/19/9/9/9/9 for 0.1,1,10,30,100,300 nM, respectively. 10 µM replotted from **C**). (**E**) To assess rapid desensitization, 3 acquisitions were performed as depicted in the inset. Cells were perfused for 10 min in the continuous presence of DAMGO 10 µM between stimulations. Paired measurements are shown for cells pretreated with DAMGO 10 µM for 18 hr before acquisition (chronic, n=12 acquisitions), and cells pretreated with β-CNA (50 nM for 5 min) before acquisition (β-CNA, n=9 acquisitions). (**F**) Time course of MOR mediated presynaptic inhibition for cells incubated with DAMGO 10 µM (n=20/11/10/11/9/13 acquisitions for 0/1/2/4/8/18 hr, respectively. Time zero and 18 hr replotted from C) or cells pretreated with β-CNA (50 nM for 5 min) before incubation with DAMGO (n=9/7/6 acquisitions for 0/2/4 hr, respectively. Time zero replotted from t=2 min in F). (**G**) Cumulative frequency curves of the normalized fluorescence at individual synapses for SEP-MOR signal (left panel) and synaptophysin-mCherry (syp-mCh, right panel) for naïve cells (n=3520 synapses) or cells pretreated with DAMGO 10 µM for 18 hr (n=3053 synapses). Note the left shift for SEP-MOR fluorescence after pretreatment indicating a loss of surface receptors. Syp-mCh fluorescence remains similar between conditions, reflecting appropriate sampling of the expression levels of recombinant fluorescent protein among synapses. *, **, *** represent p<0.05, 0.01, 0.001, respectively. See also *Figure 1—source data 1*.

The online version of this article includes the following source data for figure 1:

**Source data 1.** Source data for results graphed in *Figure 1*.

the response (*Figure 1E*, right, mean inhibition at 2 min 65.79 ± 3.41%, mean inhibition at 12 min 62.23 ± 3.39%, paired t-test, p=0.35). To directly test if long-term agonist exposure induces a depletion of surface receptors in axons, we imaged SEP-tagged MOR and quantified the fluorescence over thousands of synapses over multiple microscopic fields (*Figure 1G*). This analysis revealed that prolonged DAMGO exposure indeed reduces the presynaptic surface MOR pool (*Figure 1G*, MOR no DAMGO mean normalized fluorescence 1.60, 95% confidence interval 1.54–1.66, MOR +DAMGO 18 hours mean normalized fluorescence 1.17, 95% CI 1.08–1.27, two samples Kolmogorov-Smirnov test p<1e$^{-5}$). Together, these results indicate that presynaptic MOR tolerance is a process distinct from rapid desensitization and likely mediated by a net reduction of surface receptors on axons.

## Endocytosis, tolerance, and surface receptor depletion require MOR C-tail phosphorylation

We have previously shown that MOR undergoes rapid agonist-induced endocytosis in terminals and accumulates in endosomes, located both in terminals and in the adjacent axon shaft, which are marked by retromer complex associated with their limiting membrane (*Jullié et al., 2020*). We verified this by labeling surface MOR in axons and monitoring DAMGO-induced redistribution of surface-labeled MOR to endosomes marked by GFP-tagged VPS29, a core retromer component (*Figure 2A*, *Figure 2—video 1*). Application of DAMGO produced a significant, time-dependent accumulation of surface-labeled MOR in retromer-marked endosomes over several minutes (*Figure 2B*, repeated measure ANOVA p=0.0048).

Rapid endocytosis of MOR in axons is known to require phosphorylation of Ser and Thr residues in the MOR cytoplasmic tail (*Jullié et al., 2020*). Using the same assay, we verified that mutation of all Ser and Thr residues in the MOR tail (MOR S/T to A) abolished rapid internalization (*Figure 2C*, DAMGO compared to vehicle, repeated measure ANOVA p=0.57). MOR S/T to A strongly inhibited synaptic vesicle exocytosis acutely but mutation of all phosphorylation sites prevented the development of

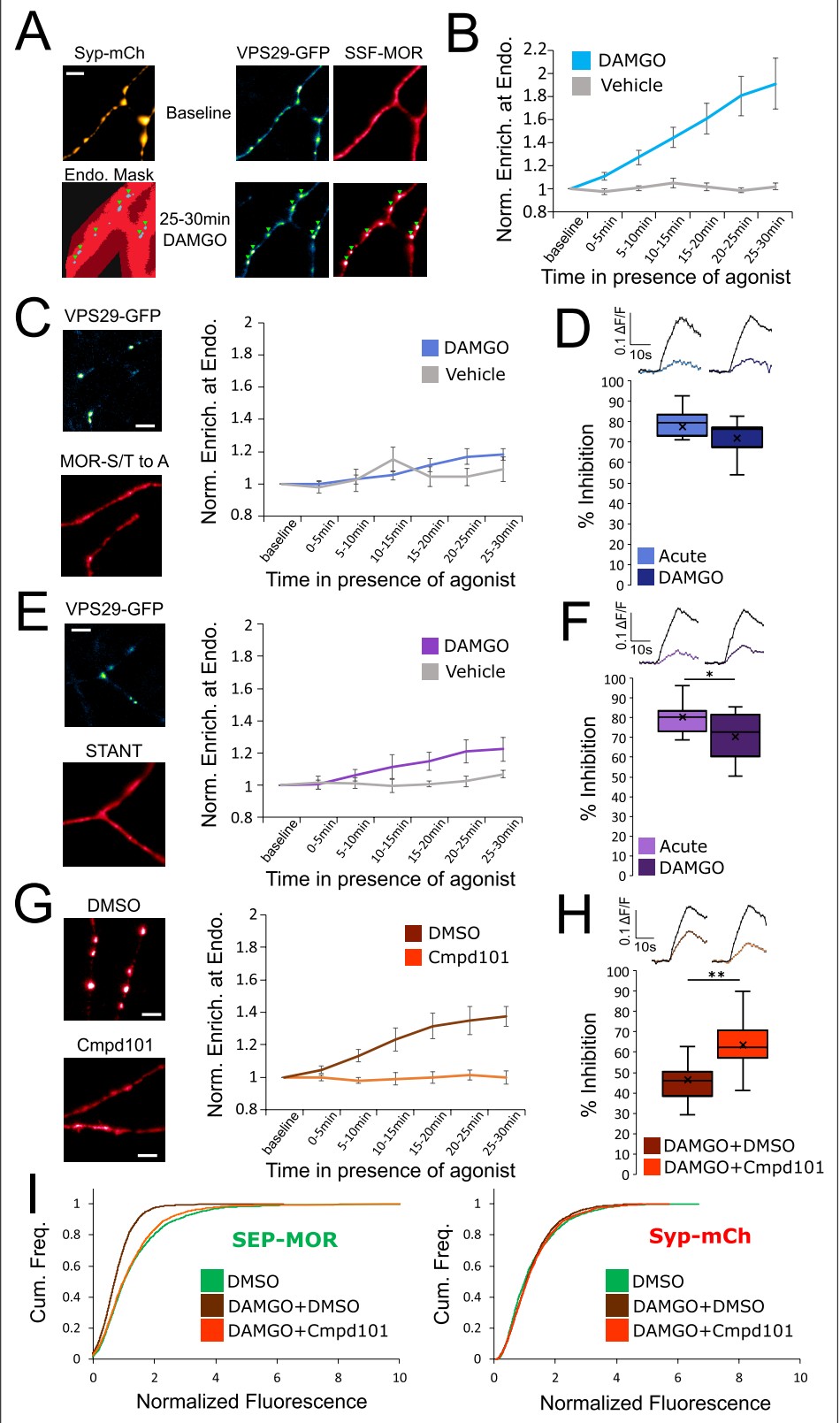

**Figure 2.** Phosphorylation of MOR is required for endocytosis of receptors, loss of surface receptors upon chronic activation, and the development of presynaptic tolerance. (**A**) Representative images of axons of neurons marked with syp-mCh, expressing the endosomal marker VPS29-GFP andFLAG-tagged opioid receptors (SSF-MOR), surface labeled with a primary anti-FLAG antibody conjugated to Alexa 647. Neurons were imaged using oblique

*Figure 2 continued on next page*

*Figure 2 continued*

illumination at a frequency of 1 frame/min. Note the uniform distribution of the receptor before agonist addition (baseline) and the punctate distribution overlapping with a segmented mask of the endosomal marker after 25–30 min of incubation with DAMGO 10 µM. Scale bar is 5 µm. See also *Figure 2—video 1*. (**B**) Quantification of the enrichment of surface labeled SSF-MOR at VPS29-GFP marked structures along axons for cells treated with vehicle (n=5 acquisitions) or cells treated with DAMGO 10 µM (n=8 acquisitions). Right axis indicates p-values for unpaired t-test between the two conditions. (**C**) Same as A,B, for FLAG-tagged mutant opioid receptors where all serine and threonine residues of the C-terminal tail have been mutated to alanine (MOR S/T to A), for vehicle (n=5 acquisitions) or DAMGO 10 µM (n=7 acquisitions) treated cells. Note the diffused distribution of surface labeled MOR-S/T to A after 25–30 min of incubation with DAMGO 10 µM. (**D**) Quantification of presynaptic inhibition mediated by the MOR S/T to A mutant acutely (inset n=717 synapses, n=7 acquisitions) or after 18 hr of incubation with DAMGO 10 µM (inset n=2360 synapses, n=11 acquisitions). (**E**) Same as A,B, for FLAG-tagged mutant opioid receptors where serine and threonine residues of STANT motif on the C-terminal tail have been mutated to alanine (STANT), for vehicle (n=5 acquisitions) or DAMGO 10 µM treated cells (n=6 acquisitions). Note the diffused distribution of surface labeled STANT after 25–30 min of incubation with DAMGO 10 µM. (**F**) Quantification of presynaptic inhibition mediated by the STANT MOR mutant acutely (inset n=1,882 synapses, n=11 acquisitions) or after 18 hr of incubation with DAMGO 10 µM (inset n=2605 synapses, n=14 acquisitions). (**G**) Same as A,B, for SSF-MOR in neurons treated with Cmpd101 30 µM (n=8 acquisitions) or DMSO control (n=6 acquisitions) and incubated with DAMGO 10 µM. Note the difference in distribution between the two conditions after 25–30 min of incubation with DAMGO. (**H**) Quantification of presynaptic inhibition mediated by wild type MOR in cells incubated with Cmpd101 30 µM (inset n=2157 synapses, n=15 acquisitions) or DMSO control (inset n=2346 synapses, n=17 acquisitions) together with DAMGO 10 µM for 18 hr. (**I**) Cumulative frequency curves of the normalized fluorescence at individual synapses for SEP-MOR signal (left panel) and syp-mCh for cells incubated with DMSO only (n=2273 synapses), cells pretreated with DMSO +DAMGO 10 µM for 18 hr (n=2209 synapses) and cells treated with Cmpd101 30 µM+DAMGO 10 µM for 18 hr (n=2456 synapses). Note that the left shift for SEP-MOR fluorescence after pretreatment with DMSO control +DAMGO is blocked by Cmpd101 while syp-mCh control signal is stable across conditions. *, ** represent p<0.05, 0.01, respectively. See also *Figure 2—source data 1*.

The online version of this article includes the following video and source data for figure 2:

**Source data 1.** Source data for results graphed in *Figure 2*.

**Figure 2—video 1.** SSF-MOR internalization in axons.

https://elifesciences.org/articles/81298/figures#fig2video1

presynaptic tolerance after chronic treatment with DAMGO (*Figure 2D*, unpaired t-test p=0.29). Key residues that regulate phosphorylation-dependent endocytosis of MOR in other systems are localized into a cluster within the C-terminal tail called the STANT motif (*Arttamangkul et al., 2019*; *Just et al., 2013*; *Lau et al., 2011*). Consistent with this, mutation of the 3 phosphorylatable residues sites in this motif strongly inhibited endocytosis of presynaptic receptors (*Figure 2E*, DAMGO compared to vehicle, repeated measure ANOVA p=0.17). The STANT mutant potently inhibited presynaptic activity acutely and little tolerance was observed after 18 hr of treatment with DAMGO (*Figure 2F*, unpaired t-test p=0.033). It is known that the GPCR kinases 2 and 3 (GRK2/3) are key regulators of MOR phosphorylation and endocytosis (*Jullié et al., 2020*; *Leff et al., 2020*; *Lowe and Bailey, 2015*; *Møller et al., 2020*). Consistent with this, compound 101 (Cmpd101), a pharmacological inhibitor of GRK2/3 activity, blocked wild-type MOR accumulation in retromer marked endosomes (Cmpd101 compared to DMSO vehicle, repeated measure ANOVA p=0.0018). Further, Cmpd101 significantly blocked the development of tolerance, verifying that phosphorylation is required for the attenuation of presynaptic MOR signaling under conditions of chronic activation (*Figure 2H*, unpaired t-test p=0.0012). Cmpd101 also blocked the loss of surface receptors induced by chronic treatment of neurons with DAMGO (*Figure 2I*, two samples Kolmogorov-Smirnov test: DMSO only mean normalized fluorescence 1.33, 95% CI 1.28–1.38, compared to DMSO +DAMGO mean normalized fluorescence 0.77, 95% CI 0.74–0.79, p<1e$^{-5}$. DMSO +DAMGO compared to Cmpd101 +DAMGO mean normalized fluorescence 1.23, 95% CI 1.19–1.28 p<1e$^{-5}$. DMSO only compared to Cmpd101 +DAMGO p=0.051). Together, these results suggest that GRK2/3-dependent phosphorylation of the MOR tail, by driving the rapid endocytosis of receptors, initiates the process of presynaptic tolerance by reducing the density of receptors present on the axon surface under conditions of prolonged opioid exposure.

## Insight to differences in the effects of chemically distinct opioid agonist drugs

DAMGO efficiently promotes phosphorylation of the MOR tail, and this is a key determinant of β-arrestin recruitment driving subsequent receptor endocytosis. Non-peptide partial agonists such as morphine are less efficacious than DAMGO for promoting receptor phosphorylation as well as endocytosis (*Just et al., 2013*; *Lau et al., 2011*). Morphine has been shown to induce presynaptic tolerance in chronically treated animals (*Fyfe et al., 2010*) but, to our knowledge, its effect on surface MOR availability on axons has not been tested. We were unable to detect significant rapid internalization of MOR in axons, measured after 30 min of morphine exposure, using our endosomal recruitment assay (*Figure 3A*, repeated measure ANOVA p=0.36). However, significant functional tolerance was detected after prolonged (18 hr) morphine exposure (*Figure 3B*, unpaired t-test, acute compared to morphine +DMSO p=0.0018). Morphine induced presynaptic tolerance to a reduced degree relative to DAMGO, but it remained dependent on GRK2/3-mediated phosphorylation because it was blocked by Cmpd101 (*Figure 3B*, unpaired t-test, morphine +DMSO compared to morphine +Cmpd101 p=0.0016). Accordingly, and despite morphine not producing detectable rapid internalization in axons, we asked whether chronic exposure to morphine is also associated with a phosphorylation-dependent reduction of the overall density of MOR on the axon surface. To test this, we imaged SEP-MOR at synapses after chronic treatment with morphine +Cmpd101 or morphine +DMSO. We found that morphine +DMSO vehicle significantly reduced the amount of receptors at the surface of axons (morphine +DMSO mean normalized fluorescence 1.01, 95% CI 0.96–1.06, compared to DMSO only, two samples Kolmogorov-Smirnov test $p<1e^{-5}$), but this effect was not as pronounced as when neurons were treated with DAMGO +DMSO (*Figure 3C*, two samples Kolmogorov-Smirnov test $p<1e^{-5}$). Furthermore, the morphine-induced reduction of surface receptor number was inhibited by Cmpd101 (*Figure 3C*, morphine +Cmpd101 mean normalized fluorescence 1.23, 95% CI 1.18–1.27, compared to morphine +DMSO two samples Kolmogorov-Smirnov test $p<1e^{-5}$). These observations suggest that morphine, despite promoting MOR endocytosis only weakly compared to DAMGO, is indeed able to produce presynaptic tolerance after chronic exposure through a similar phosphorylation-dependent mechanism.

G protein-biased agonists are thought to stimulate MOR internalization even less strongly than morphine. We therefore tested two such compounds, PZM21 and TRV130 (*DeWire et al., 2013*; *Manglik et al., 2016*). We could not detect any significant internalization induced by bath application of PZM21 (repeated measure ANOVA, *P*=0.68), nor tolerance to DAMGO mediated presynaptic inhibition after 18 hr of incubation with the biased agonist (*Figure 3D and E*, unpaired t-test, acute compared to PZM21 +DMSO p=0.37, PZM21 +DMSO compared to PZM21 +Cmpd101 p=0.30). Similarly, TRV130 failed to produce significant MOR internalization (repeated measure ANOVA p=0.39) or tolerance (*Figure 3F and G*, unpaired t-test, acute compared to TRV130 +DMSO p=0.37, TRV130 +DMSO compared to TRV130 +Cmpd101 p=0.091). Together, these results establish a positive correlation between the endocytic efficacy of chemically diverse MOR agonists and the observed degree of tolerance that they produce.

## DOR exhibits a higher degree of tolerance than MOR and indicates that tolerance is an homologous process

MOR is not the only receptor mediating presynaptic neuromodulation by opioids. DOR is another well-known example that mediates Gi-coupled inhibition of neurotransmitter release in response to opioids (*Bardoni et al., 2014*; *He et al., 2021*; *Piskorowski and Chevaleyre, 2013*). Physiological tolerance to DOR-mediated effects is well established (*DiCello et al., 2019*; *Pradhan et al., 2010*), and agonist-induced internalization of DOR has been clearly demonstrated in the soma and dendrites of neurons (*Pradhan et al., 2009*; *Scherrer et al., 2006*). Recent evidence indicates that DOR does not rapidly desensitize at the presynapse (*He et al., 2021*). However, DOR trafficking in axons has not been studied previously, and it is not known if longer-term tolerance develops to DOR-mediated presynaptic inhibition. We found that, similar to MOR, surface labeled DOR is diffusely distributed in axons of striatal neurons and does not detectably accumulate at synapses marked with syp-mCh under basal conditions (*Figure 4A*, baseline). After stimulation of DOR with the peptide agonist [D-Ala$^2$, D-Leu$^5$]-Enkephalin (DADLE), there was a redistribution of surface receptors in punctate structures that colocalized with the retromer marker VPS29-GFP (*Figure 4A and B*, *Figure 4—video 1*.

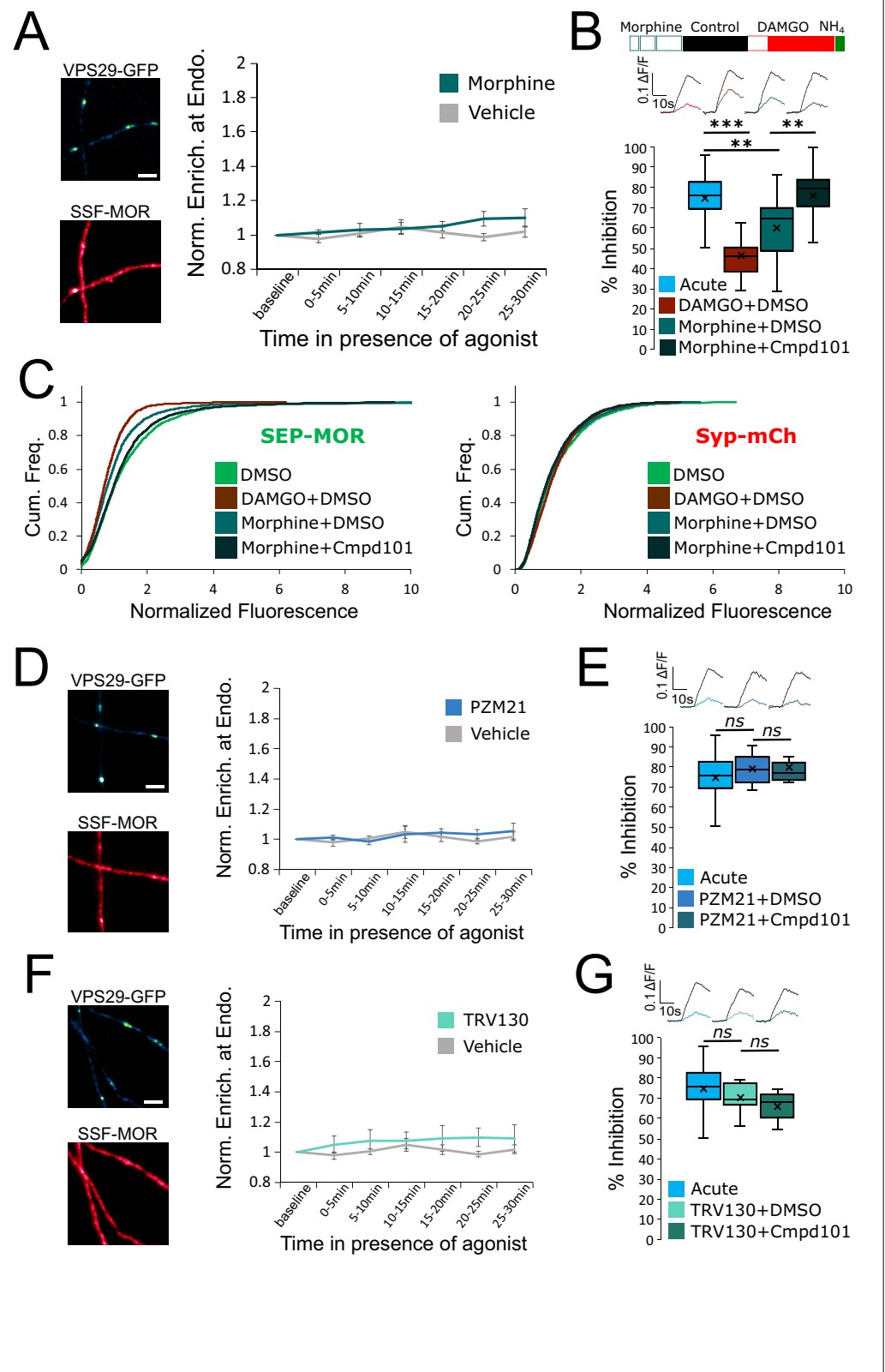

**Figure 3.** Partial and biased MOR agonists fail to elicit tolerance to the same degree as full agonists peptides. (**A**) Same experimental setup as in *Figure 2*, except the agonist used was morphine 10 µM (n=8 acquisitions), control replotted from *Figure 2B*. Inset show images of VPS29-GFP and surface labeled SSF-MOR after 25–30 min of incubation with morphine 10 µM. (**B**) DAMGO induced MOR inhibition exhibits tolerance after incubation

*Figure 3 continued on next page*

*Figure 3 continued*

with morphine 10 µM+DMSO vehicle (inset n=3,400 synapses, n=21 acquisitions) and tolerance is blocked by incubation of morphine 10 µM together with Cmpd101 30 µM (inset n=2441 synapses, n=19 acquisitions). Acute condition replotted from *Figure 1C*, DMSO +DAMGO condition replotted from *Figure 2H*. (**C**) Morphine 10 µM+DMSO (n=2076 synapses) induces a loss of surface SEP-MOR in axons after 18 hr of incubation compared to DMSO only control (replotted from *Figure 2I*). The loss is less pronounced than when induced by incubation by DAMGO 10 µM+DMSO (replotted from *Figure 2I*) for 18 hr, and is blocked by incubation of DAMGO 10 µM together with Cmpd101 30 µM (n=2872 synapses). Syp-mCh signal is similar across conditions. (**D**) Same as **A** except cells were stimulated with PZM21 10 µM (n=5 acquisitions). Note the diffuse distribution of surface labeled SSF-MOR after 25–30 min of incubation with PZM21 10 µM. (**E**) Same as for **B** for cells incubated for 18 hours with PZM21 10 µM together with Cmpd101 30 µM for 18 hr (inset n=883 synapses, n=8 acquisitions) or DMSO vehicle (inset n=1089 synapses, n=8 acquisitions). Acute condition replotted from *Figure 1C*. (**F**) Same as **A** except cells were stimulated with TRV130 10 µM (n=5 acquisitions). Note the diffuse distribution of surface labeled SSF-MOR after 25–30 min of incubation with TRV130 10 µM. (**G**) Same as for **B** for cells incubated for 18 hr with TRV130 10 µM together with Cmpd101 30 µM for 18 hr (inset n=1,076 synapses, n=7 acquisitions) or DMSO vehicle (inset n=1280 synapses, n=8 acquisitions). Acute condition replotted from *Figure 1C*. Scale bars are 5 µm. **, *** represent p<0.01, 0.001, respectively. See also *Figure 3—source data 1*.

The online version of this article includes the following source data for figure 3:

**Source data 1.** Source data for results graphed in *Figure 3*.

DADLE compared to vehicle repeated measure ANOVA, p=0.076). This indicates that, as for MOR, presynaptic DOR undergoes ligand dependent endocytosis and accumulates in a similar population of presynaptic endosomes. Using our optical assay to probe presynaptic inhibition, we found that DADLE-induced activation of DOR produces a potent inhibition of synaptic vesicle exocytosis (*Figure 4C and D*). We could not detect significant attenuation of this response after 10 min of agonist application, suggesting that presynaptic inhibition mediated by DOR is resistant to acute desensitization (*Figure 1C*, mean inhibition at 2 min 81.90 ± 2.76%, mean inhibition at 12 min 82.96 ± 4.55%, paired t-test, p=0.80). We probed presynaptic DOR tolerance by measuring inhibition after continuous agonist exposure for 18 hr. DOR-mediated inhibition of synaptic vesicle exocytosis was barely detectable after this chronic treatment, indicating robust tolerance (*Figure 4D*, unpaired t-test, acute compared to chronic $p<1e^{-5}$). Accordingly, while both DOR and MOR -mediated presynaptic inhibition are resistant to acute desensitization yet become tolerant after chronic agonist exposure, the degree of tolerance development is greater degree for DOR when assessed under comparable conditions (*Figure 4D*, unpaired t-test, MOR compared to DOR after chronic treatment $p=1.01e^{-5}$).

DOR is co-expressed with MOR in some neurons, and it has been proposed that such co-expression can underlie functional cross-talk and cross-regulation between these distinct opioid receptor types (*He et al., 2021*; *Wang et al., 2018*). This motivated us to ask if presynaptic tolerance elicited by chronic agonist exposure is receptor-specific or if chronic activation of one receptor type promotes the development of tolerance at the other. Neurons co-expressing MOR and DOR were probed in our optical assay after chronic treatment with either the MOR-selective full agonist DAMGO or the DOR-selective full agonist [D-Pen[2,5]]-Enkephalin (DPDPE). MOR tolerance was induced by DAMGO but not by DPDPE (*Figure 4E*, mean MOR inhibition after DAMGO chronic 45.85 ± 4.19%, mean MOR inhibition after DPDPE chronic 66.09 ± 4.80%, unpaired t-test p=0.0039). Conversely, DOR tolerance was induced by DPDPE but not by DAMGO (mean DOR inhibition after DAMGO chronic 59.10 ± 4.59%, mean DOR inhibition after DPDPE chronic 39.72 ± 5.58%, unpaired t-test p=0.012). Together, these results indicate that both DOR and MOR mediate presynaptic inhibition when co-expressed at the same terminals and that both responses develop significant tolerance after chronic agonist exposure. However, tolerance to each opioid response is induced in a homologous manner, indicating that its development is receptor-specific.

## Differences in the degree of presynaptic tolerance between receptor types correlate with differences in surface receptor depletion and recycling rate

As our results establish a link between the development of presynaptic tolerance and a reduction in the surface pool of opioid receptors, we anticipated from the above results that loss of surface

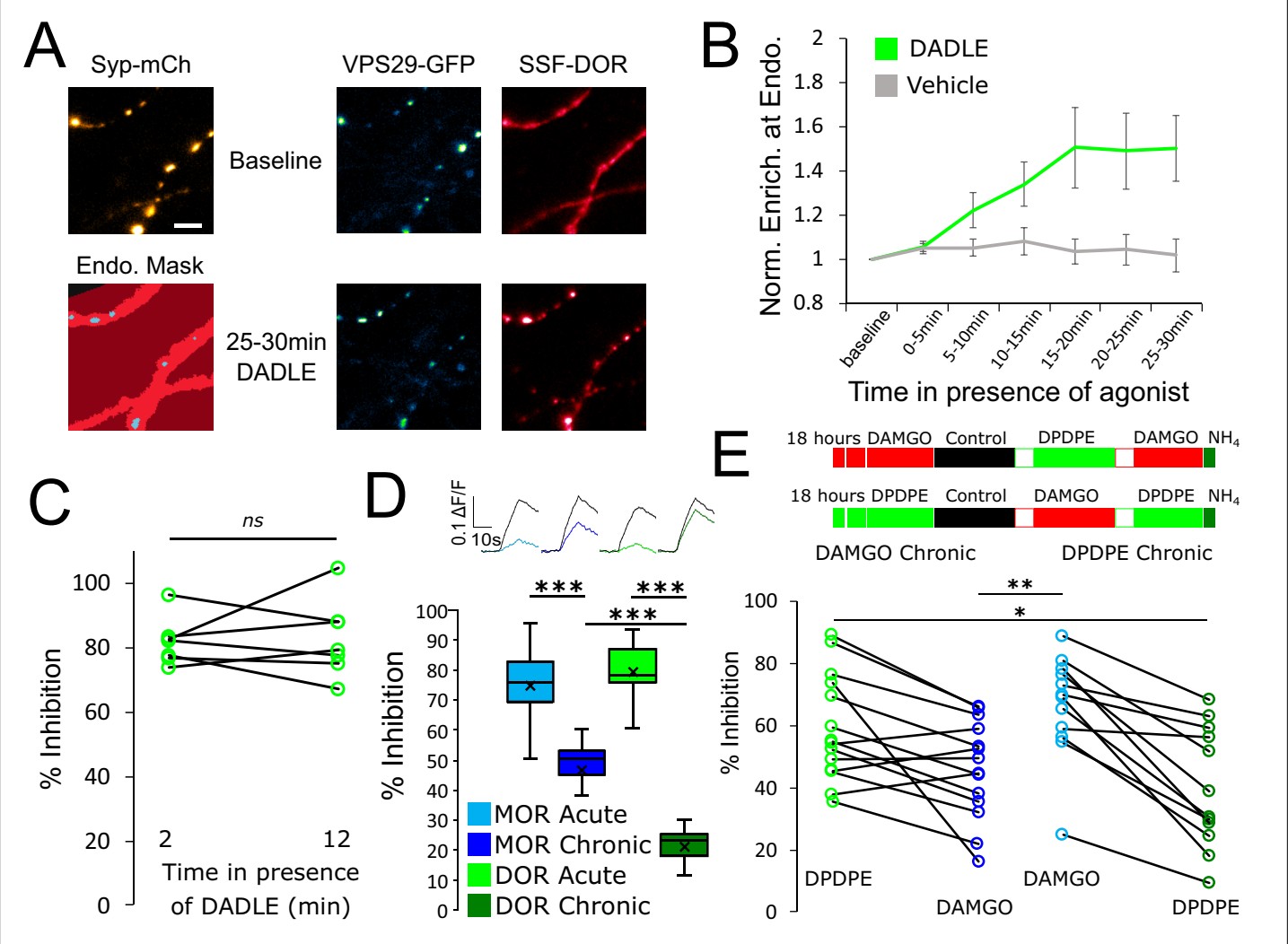

**Figure 4.** Tolerance is an homologous process conserved between opioid receptors. (**A**) Representative images of axons of neurons marked with syp-mCh, expressing the endosomal marker VPS29-GFP and FLAG-tagged DOR (SSF-DOR) surface labeled with a primary anti-FLAG antibody conjugated to alexa647. Imaging was performed as described before. Note the uniform distribution of surface SSF-DOR before agonist addition (baseline) and the punctate distribution overlapping with a segmented mask of the endosomal marker after 25–30 min of incubation with DADLE 10 μM. Scale bar is 5 μm. See also *Figure 4—video 1*. (**B**) Time course of surface labeled SSF-DOR recruitment at VPS29-GFP marked presynaptic endosomes, as in A. There is a significant increase in colocalization of SSF-DOR with the retromer marker after addition of DADLE 10 μM (n=11 acquisitions) compared to the vehicle control (n=7 acquisitions). (**C**) Inhibition of electrically evoked exocytosis of synaptic vesicles by DOR is sustained over 10 min in presence of agonist. Desensitization of presynaptic DOR was assessed using a similar protocol as in *Figure 1E*. Neurons expressing SSF-DOR and VAMP2-SEP were electrically stimulated to induce SV exocytosis in control solution. Cells were then perfused with a solution containing DADLE 10 μM and inhibition of the fluorescence increase was quantified to reflect DOR mediated presynaptic inhibition. After 10 more minutes of perfusion with DADLE, cells were stimulated again to estimate the degree of acute desensitization (n=7 acquisitions). (**D**) Quantification of the presynaptic inhibition mediated by MOR in acute and chronic conditions (replotted from *Figure 1C*) compared to DOR, acutely (inset n=1,482 synapses, n=10 acquisitions) or after 18 hr of treatment with DADLE 10 μM (inset n=2,529 synapses, n=11 acquisitions). (**E**) Assessment of cross-tolerance using optical measurement of presynaptic inhibition. Inset describes experimental setup. Neurons expressing VAMP2-SEP together with SSF-DOR and SSF-MOR were incubated for 18 hr with either DAMGO 10 μM (n=14 acquisitions) or DPDPE 10 μM (n=12 acquisitions). Cells treated chronically with DAMGO were electrically stimulated while imaged in control solution, then 2 min after perfusion with 10 μM DPDPE, then 2 min after exchange for a solution containing 10 μM DAMGO. Cells treated chronically with DPDPE were submitted to the same protocol, except that the order of solution perfusion was reversed (DAMGO first, DPDPE second). *, **, *** represent p<0.05, 0.01, 0.001, respectively. See also *Figure 4—source data 1*.

The online version of this article includes the following video and source data for figure 4:

**Source data 1.** Source data for results graphed in *Figure 4*.

**Figure 4—video 1.** SSF-DOR internalization in axons.

https://elifesciences.org/articles/81298/figures#fig4video1

receptors occurs to a greater degree for DOR than MOR. To test this, we imaged surface DOR fluorescence using a N-terminally SEP-tagged construct. We found that 18 hr of incubation with DADLE led to reduction in the number of surface receptors, indicating that DOR tolerance is paralleled by a reduction of the receptor pool in axons (DOR no DADLE mean normalized fluorescence 1.43, 95% CI 1.38–1.48, compared to DOR +18 hr DADLE mean normalized fluorescence 0.32, 95% CI 0.31–0.34, two samples Kolmogorov-Smirnov test $p < 1e^{-5}$). Also, the loss of receptor induced by chronic treatment was much more pronounced than for SEP-MOR, in agreement with our measurements of presynaptic inhibition after induction of tolerance (*Figure 5A*, MOR +DAMGO compared to DOR +DADLE, two samples Kolmogorov-Smirnov test $p < 1e^{-5}$). Enhanced down-regulation of surface DOR relative to MOR has been observed previously in non-neural models, and it results from a reduced efficiency of DOR to enter the recycling pathway compared to MOR (*Tanowitz and von Zastrow, 2003*). To test if this is the case in axons, we co-expressed syp-mCh together with SEP-tagged receptors and exposed neurons to agonist for 20 min, a time sufficient to strongly drive receptors into the endocytic pathway. We then imaged axons at an acquisition rate sufficiently high (10 Hz) to resolve individual receptor-containing vesicular fusion events mediating receptor recycling to the axon surface; these appear as bursts of fluorescence due to rapid SEP dequenching upon exposure to the neutral extracellular milieu (*Figure 5B*). Such insertion events were detected for both SEP-DOR and SEP-MOR, with no significant difference in amplitude (*Figure 5C*, unpaired t-test, p=0.93). While recycling events were rare for both receptors in neurons not pretreated with agonist, their frequency was significantly higher after agonist treatment consistent with ligand-induced trafficking and recycling (unpaired t-test, MOR p=0.00023, DOR p=0.0015). More importantly, we found that after agonist treatment, the frequency of SEP-DOR surface insertion events was about half of what was observed for SEP-MOR (unpaired t-test, p=0.0231). Together, these data suggest that, while both MOR and DOR undergo robust ligand-dependent endocytosis in axons, DOR recycles less efficiently than MOR. We propose that, when iterated over the course of 18 hr of agonist treatment, this produces a difference in the degree of progressive receptor depletion from the axon surface which underlies the observed difference in magnitude of functional tolerance development.

## Presynaptic DOR endocytosis, surface receptor depletion and tolerance do not require C-tail phosphorylation

Whereas both MOR and DOR appear to rely on endocytosis-dependent loss of axonal receptors for the development of presynaptic tolerance, we found the biochemical requirements for this control to be remarkably different between the two opioid receptor types. Specifically, while MOR internalization, surface receptor loss, and tolerance clearly require phosphorylation of the receptor's cytoplasmic tail, this was not the case for DOR. First, the degree of DADLE-induced DOR tolerance observed in the presence of Cmpd101 was indistinguishable from the DMSO vehicle control (*Figure 6A*, unpaired t-test, p=0.71). We also observed rapid internalization of DOR in the presence of Cmpd101 (*Figure 6B*, Cmpd101 compared to DMSO vehicle, repeated measure ANOVA, p=0.70). These results indicate that DOR tolerance and internalization do not require GRK2/3 activity, in contrast to MOR. Second, mutating all Ser and Thr residues in the DOR C-terminal tail (DOR S/T to A) did not prevent the development of presynaptic tolerance (*Figure 6C*, unpaired t-test, p=0.00036). We also observed significant rapid internalization of DOR S/T to A, as indicated by the mutant receptor undergoing DADLE-induced accumulation at retromer marked endosomes (*Figure 6D*, *Figure 6—video 1*, repeated measure ANOVA, *P*=0.0335). These results indicate that DOR tolerance and internalization do not require phosphorylation of the receptor's cytoplasmic tail, in contrast to MOR. Third, assay of surface receptor fluorescence indicated that the S/T to A mutation does not prevent DADLE-induced reduction of the surface pool of SEP-DOR (*Figure 6E*, DOR S/T to A no DADLE mean normalized fluorescence 1.30, 95% CI 1.27–1.32, compared to DOR S/T to A+18 hr DADLE mean normalized fluorescence 0.45, 95% CI 0.44–0.46, two samples Kolmogorov-Smirnov test $p < 1e^{-5}$). This indicates that agonist-induced reduction of the axonal surface DOR pool can occur in the complete absence of phosphorylation of the cytoplasmic tail.

Altogether our data suggest that endocytosis is responsible for long-term surface receptor depletion for both MOR and DOR, but that significant endocytosis of DOR can occur in the absence of phosphorylation of the receptor's cytoplasmic tail. To confirm this result, we used a different assay that leverages the properties of SEP. Ammonium chloride ($NH_4Cl$) can titrate acidic intracellular

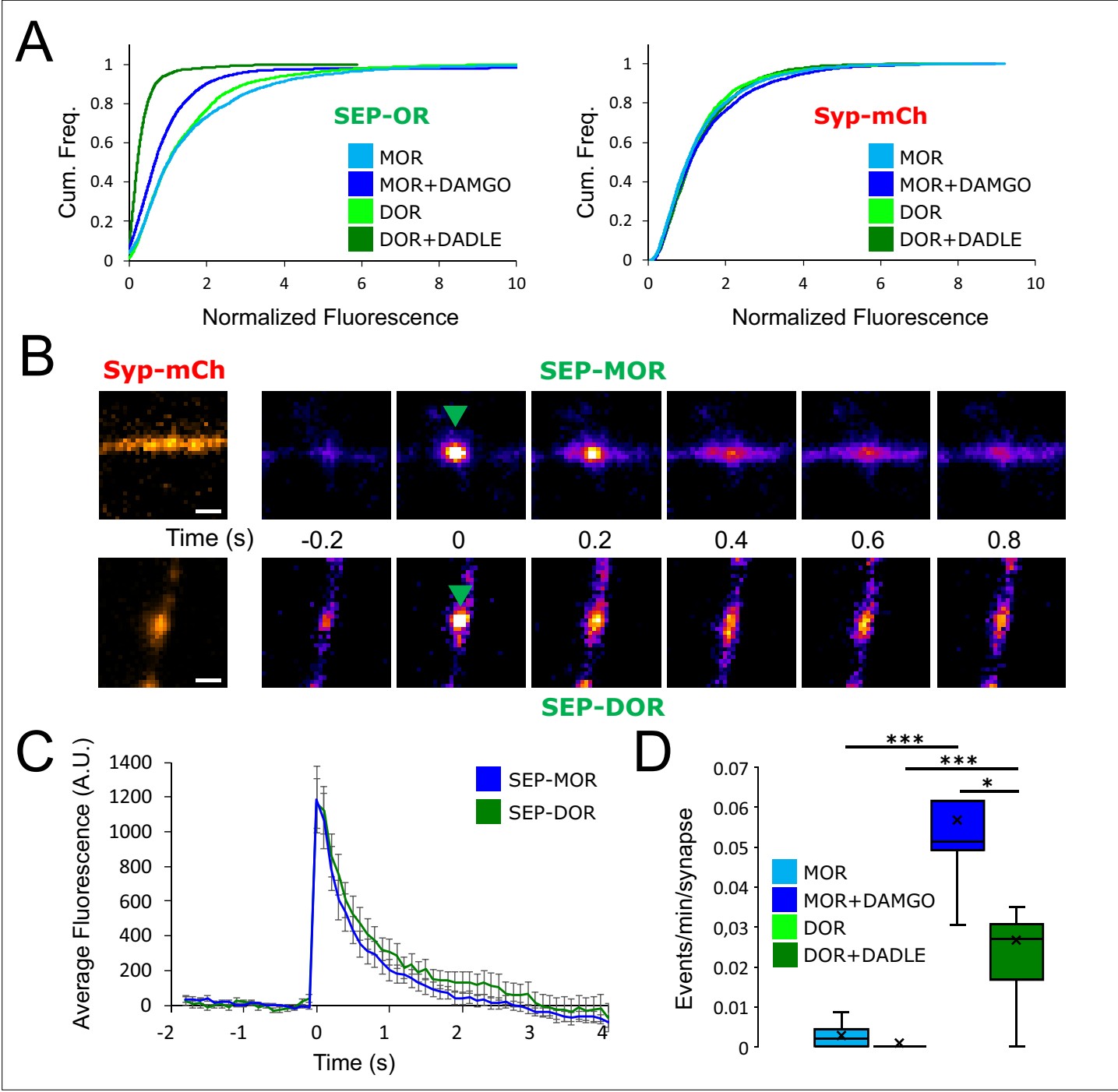

**Figure 5.** DOR is less efficiently recycled to the plasma membrane compared to MOR. (**A**) Eighteen hr of incubation with DADLE 10 μm (n=2934 synapses) induces a marked loss of surface SEP-DOR in axons compared to the untreated control (n=3,376 synapses). The loss is more pronounced than what is observed after chronic treatment of SEP-MOR with DAMGO (replotted from *Figure 1G*). Syp-mCh fluorescence control remains similar across conditions. (**B**) Representative examples of surface insertion of SEP-tagged opioid receptors. Neurons were incubated for 20 min with either DAMGO 10 μM (for SEP-MOR) or DADLE 10 μM (for SEP-DOR), and imaged at 10 Hz using oblique illumination. Insertion events appear as bursts of fluorescence (green arrow). Scale bar is 1 μm. (**C**) Average fluorescence intensity profile at the site of insertion for SEP-MOR (n=59 events) and SEP-DOR (n=49 events), for events imaged as described in B. Error bars represent SEM. (**D**) Whisker plots of the normalized frequency of surface insertion events for neurons imaged as described in B. Frequency of recycling events was increased for both MOR (n=5 acquisitions) and DOR (n=8 acquisitions) compared to the no agonist pretreatment condition (MOR n=6 acquisitions, DOR n=8 acquisitions). *, *** represent p<0.05, 0.001, respectively. See also *Figure 5—source data 1*.

The online version of this article includes the following source data for figure 5:

*Figure 5 continued on next page*

*Figure 5 continued*

**Source data 1.** Source data for results graphed in *Figure 5*.

compartments and reveal SEP fluorescence from internal receptors. In neurons expressing SEP-DOR S/T to A, and in basal conditions, application of NH$_4$Cl induced a modest fluorescence increase (Δ1, mean = 7.39 ± 0.75%), suggesting that SEP-DOR S/T to A mostly resides at the surface of the axon. After 30 min of DADLE bath application, NH$_4$Cl application led to a significantly larger fluorescence increase (Δ2, mean = 16.45 ± 2.27%, paired t-test p=0.0017), indicating that a proportion of mutant receptors had relocalized from the surface to internal acidic organelles (*Figure 6F and G*). These results independently confirm that phosphorylation of the DOR tail is not essential for endocytosis in axons.

## Discussion

The present results establish that GRK-mediated phosphorylation of the MOR cytoplasmic tail drives a form of cellular opioid tolerance at the presynaptic terminal which develops over a significantly longer period of time than the previously elucidated process of rapid desensitization of postsynaptic MOR signaling. Our results support a simple cellular mechanism sufficient to explain the slower kinetics of presynaptic tolerance development, based on an iterative endocytic trafficking cycle that mediates progressive depletion of receptors from the axon surface in the presence of chronic agonist exposure. Accordingly, the present results provide new insight into the cellular and molecular basis for differences in the timescales over which functionally relevant neuroadaptations to opioids develop.

Presynaptic tolerance develops gradually because it represents an integrated effect of repetitive rounds of endocytosis and recycling, in which a limited fraction of internalized receptors are not reinserted in each cycle (*Figure 7*, large arrow). The key event initiating this iterative trafficking cycle is agonist-induced endocytosis of the receptor and, for MOR, this requires phosphorylation of the receptor's cytoplasmic tail. Supporting this conclusion, blocking MOR phosphorylation in multiple ways also prevents the gradual depletion of surface receptors and the development of functional presynaptic tolerance. Further, agonists which do not drive MOR endocytosis robustly produce less (morphine) or no (PZM21, TRV130) measureable presynaptic tolerance. The idea that tolerance develops as a consequence of gradual depletion of the total receptor pool on the axon surface is also consistent with our finding that β-CNA, a distinct manipulation which reduces total receptor reserve, dramatically accelerates the development of opioid-induced presynaptic tolerance. Thus, the present model is supported at multiple levels and is sufficient to explain the extended time course over which presynaptic opioid tolerance develops.

A limitation of the present study is that it relies on the expression of recombinant opioid receptors in cultured neurons. We previously showed that our experimental system can detect presynaptic inhibition mediated through endogenous opioid receptors (*Jullié et al., 2020*). However, in the embryonic striatal cultures used for the present study, only a small proportion of neurons express opioid receptors and this dilutes the degree of endogenous inhibition when analyzed at the population level. We therefore used electroporation to co-express receptors with the VAMP2-SEP reporter. In a previous study, we showed that this method yields levels of recombinant receptor expression within the range of endogenous receptor expression observed across individual neurons in culture (*Jullié et al., 2020*). However, these levels can vary considerably, both between neurons and brain regions. Our concentration-response analysis (*Figure 1D*) suggests that the present model system has higher opioid sensitivity than slice preparations in which presynaptic inhibition by endogenous receptors was previously described (*Fyfe et al., 2010*; *Pennock and Hentges, 2011*). Accordingly, caution is advised in comparing across systems. Nevertheless, the present results suggest that endocytic trafficking is capable of producing substantial presynaptic tolerance even at receptor expression levels that exceed endogenous (which one might expect to mask tolerance due to elevated receptor reserve). The relevance of presynaptic endocytic trafficking to the development of physiological tolerance in vivo also remains to be determined. Interestingly, we note that biased agonism at MOR has been reported to produce less antinociceptive tolerance in vivo (*Altarifi et al., 2017*; *Singleton et al., 2021*), and that a measure of postsynaptic cellular tolerance appears to require phosphorylation of the MOR cytoplasmic tail (*Arttamangkul et al., 2018*).

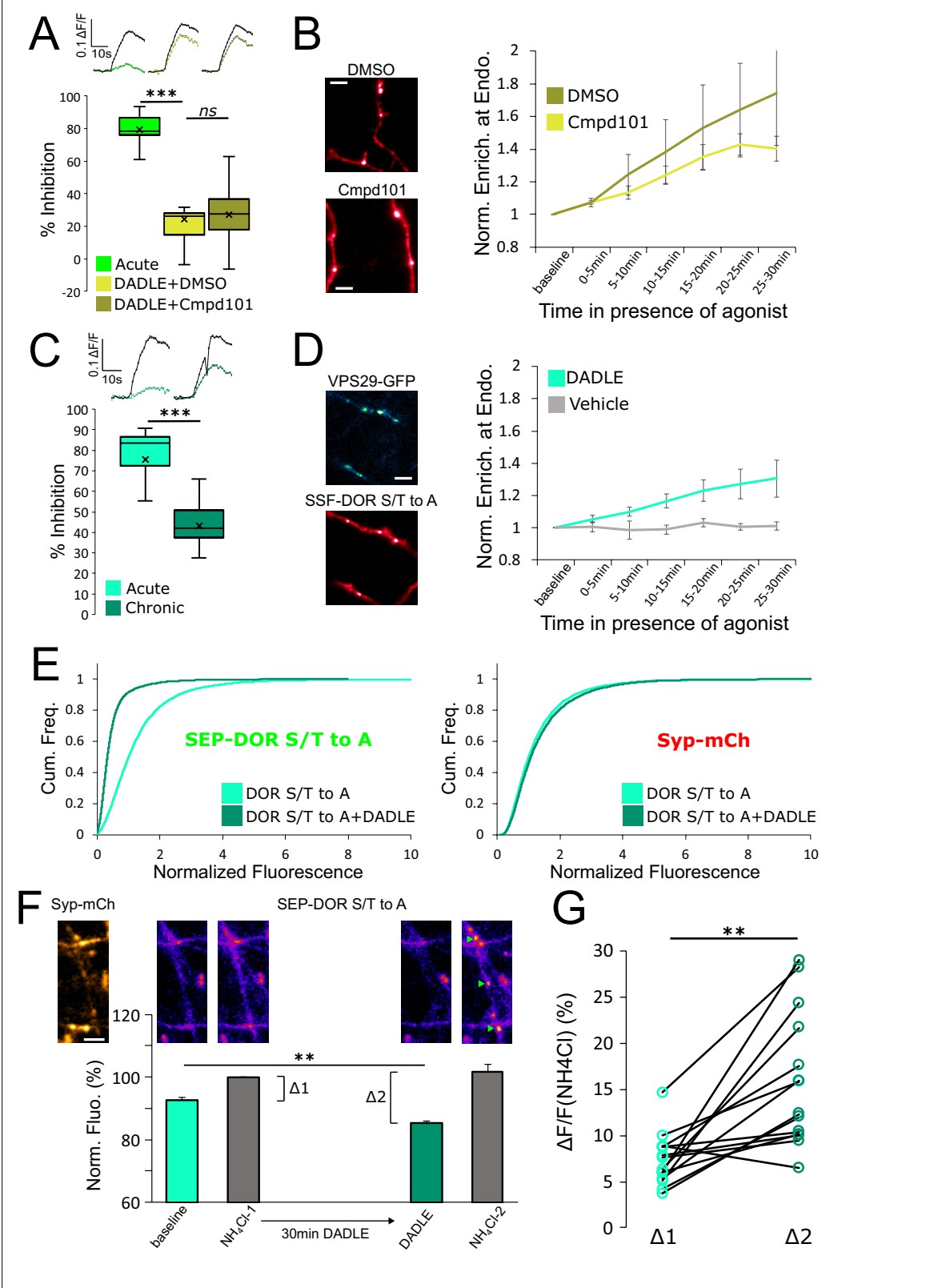

**Figure 6.** DOR C-terminal tail phosphorylation is not necessary for presynaptic endocytosis, surface receptor depletion or tolerance. (**A**) Inhibition of GRK2/3 activity does not block DOR tolerance. Neurons were incubated with Cmpd101 30 μM+DADLE 10 μM (inset n=2481 synapses, n=15 acquisitions) or DMSO vehicle +DADLE 10 μM (inset n=2353 synapses, n=13 acquisitions) for 18 hr, and inhibition of presynaptic activity measured as described previously. Acute condition is replotted from **Figure 4D**. (**B**) Effect of GRK2/3 inhibition on the accumulation of surface labeled SSF-DOR

*Figure 6 continued on next page*

*Figure 6 continued*

at VPS29-GFP marked endosomes, as previously described. Neurons were incubated with Cmpd101 30 µM (n=7 acquisitions) or DMSO vehicle (n=7 acquisitions) and DADLE 10 µM was added after baseline. Note the punctate distribution for both conditions after 25–30 min of incubation with agonist. Scale bar is 5 µm. (**C**) DOR S/T to A develops tolerance after chronic activation. Presynaptic inhibition mediated by a phosphorylation-deficient mutant of DOR was assessed in neurons treated acutely (inset n=1,673 synapses, n=8 acquisitions) or in neurons pretreated for 18 hr with DADLE (inset n=3017 synapses, n=12 acquisitions). (**D**) Endocytosis of DOR S/T to A, as described previously. Neurons expressing the mutant were stimulated with DADLE 10 µM (n=7 acquisitions) or vehicle control (n=6 acquisitions). Note the punctate distribution overlapping the endosomal marker signal after 25–30 min of incubation with agonist. Scale bar is 5 µm. See also *Figure 6—video 1*. (**E**) Normalized fluorescence of SEP-DOR S/T to A at synapses marked by syp-mCh in naive neurons (n=7,200 synapses) or in neurons pretreated with DADLE 10 µM for 18 hr (n=8904 synapses). Note the left shift in SEP-DOR S/T to A fluorescence after chronic activation while the syp-mCh signal remains stable. (**F**) Endocytosis of SEP-DOR S/T to A assessed by pHluorin unquenching. Axons were identified by syp-mCh staining and perfused with imaging solution, and a baseline image acquired. One min after perfusion of $NH_4Cl$ solution, another image was taken ($NH_4Cl$-1) showing little increase in fluorescence (Δ1). Cells were then perfused for 30 min with DADLE 10 µM in imaging solution, and another frame acquired (DADLE). One last frame was acquired after 1 min of perfusion with $NH_4Cl$ ($NH_4Cl$-2) showing a larger increase in fluorescence (Δ2). Inset shows representative images for each step, green arrows point to fluorescent punctates that represent endosomes. Scale bar is 5 µm, n=14 acquisitions. (**G**) Paired measurement of the fluorescence increase induced by $NH_4Cl$ at baseline (Δ1) or after 30 min of incubation with DADLE 10 µM (Δ2), as described in E, same dataset. **, *** represent p<0.01, 0.001, respectively. See also *Figure 6—source data 1*.

The online version of this article includes the following video and source data for figure 6:

**Source data 1.** Source data for results graphed in *Figure 6*.

**Figure 6—video 1.** SSF-DOR S/T to A internalization in axons.

https://elifesciences.org/articles/81298/figures#fig6video1

A cellular basis for the ability of presynaptic inhibition by MOR to resist rapid desensitization was proposed previously, grounded in two distinguishing properties of MOR cell biology in axons. First, endocytosis of receptors occurs almost exclusively in synaptic specializations that are sparsely distributed along the extended axon shaft. Second, opioid receptors are laterally mobile over the entire axon surface, with receptors present at synapses able to exchange with the adjacent extrasynaptic pool within seconds (*Jullié et al., 2020*). Accordingly, receptors present on the axon surface, but outside

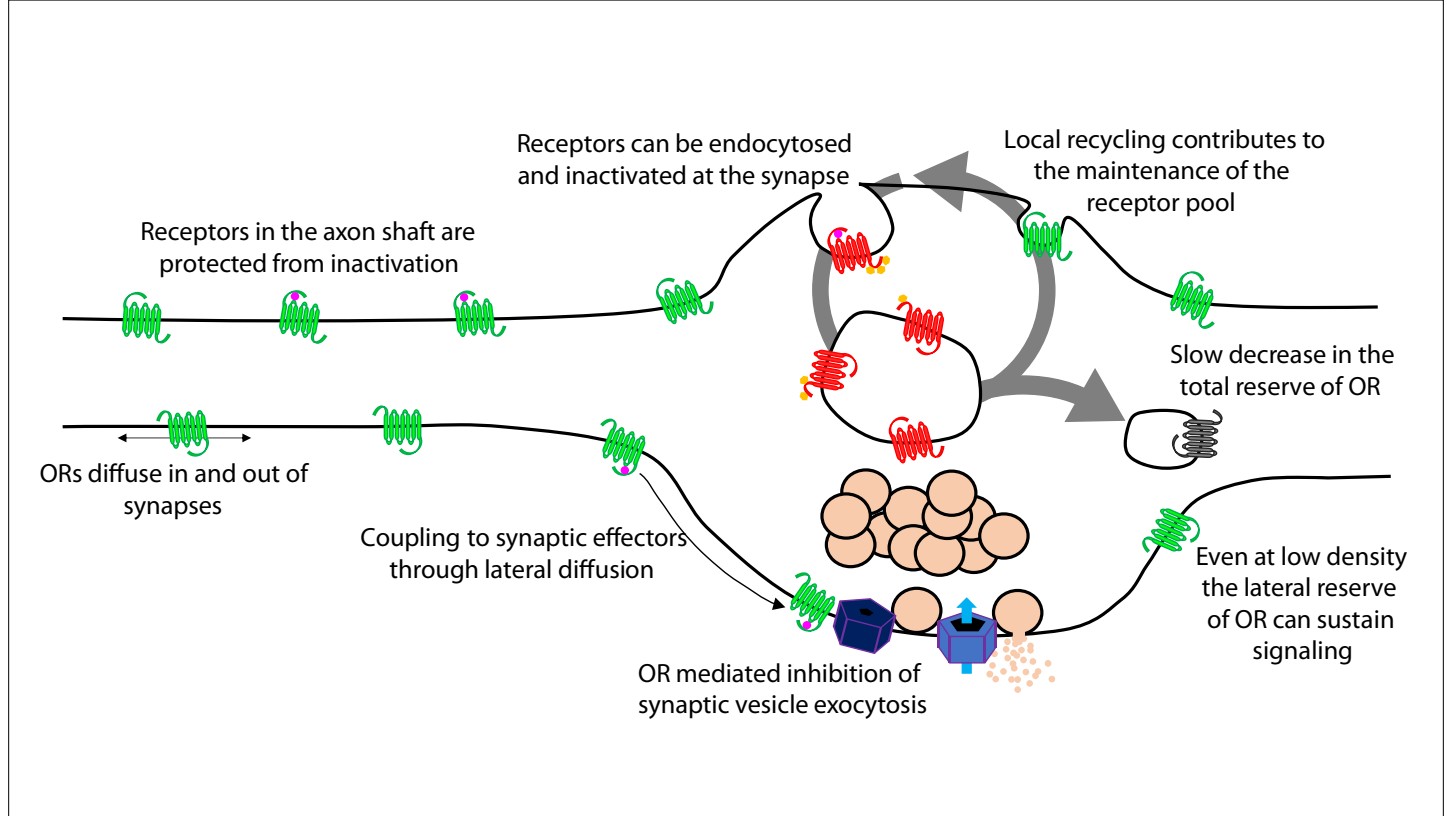

**Figure 7.** Proposed model of opioid receptor signaling and trafficking in axons.

of synapses, are able to diffuse into synapses at a faster rate than receptors undergo agonist-induced endocytic capture and inactivation within synapses. The extrasynaptic membrane thus provides an expansive 'lateral reserve' of receptors that, through lateral diffusion, enables opioid signaling at terminals to be maintained even at low overall surface receptor density (*Figure 7*, small arrows). The present results extend this general concept to DOR and suggest that presynaptic tolerance develops in a distinct manner – for both MOR and DOR – through a progressive reduction in the 'total reserve' of receptors present on the axon surface (*Figure 7*, large arrow). By elaborating the classical concept of receptor reserve into two components, it becomes possible to simply explain how presynaptic inhibition mediated by opioid receptors is able to resist rapid desensitization (maintained by the lateral reserve) while developing significant tolerance over a longer time period (due to reduction of the total reserve).

Our proposed model for the development of presynaptic tolerance is reminiscent of a general paradigm of GPCR downregulation that was pioneered through the study of DOR in non-neural cells (*Law et al., 1982*; *Williams et al., 2013*). Our results are consistent with this model, including the present demonstration that DOR recycles less efficiently than MOR in axons and develops presynaptic tolerance more robustly. A key point of divergence is that, for DOR, none of the events associated with the development of tolerance at the presynapse – beginning with endocytosis of the receptor – requires phosphorylation of the receptor tail. Agonist-induced phosphorylation of the DOR tail has been clearly demonstrated, and many studies support its importance for mediating DOR endocytosis and/or downregulation in other cellular contexts (e.g., *Mann et al., 2020*; *Whistler et al., 2001*). However, we also note that there is evidence indicating that phosphorylation is not absolutely required for DOR endocytosis (*Murray et al., 1998*; *Qiu et al., 2007*; *Zhang et al., 2005*). Important future directions include determining how this difference in the cellular regulation of MOR and DOR is mediated, and if phosphorylation-independent endocytosis of DOR is specific to neurons or the presynaptic compartment.

In sum, our results inform the fundamental question of how opioids produce neuroadaptive effects that span a wide range of timescales, and they delineate a cellular mechanism mediating the development of presynaptic tolerance to opioids. The present results are limited to two opioid receptor types. However, we note that MOR and DOR belong to the largest GPCR subclass (family A), and that presynaptic inhibition mediated by the A1 adenosine receptor (another family A GPCR) resists rapid desensitization yet develops tolerance gradually (*Wetherington and Lambert, 2002*). Thus we anticipate that the present study, in addition to providing specific insight into the neurobiology of opioids, delineates a framework and methodology useful for investigating presynaptic neuromodulation through GPCRs more generally.

## Materials and methods
### Primary rat striatal neuron cultures

All procedures were performed according to the National Institutes of Health Guide for Care and Use of Laboratory Animals and approved by the University of California San Francisco Institutional Animal Care and Use Committee (protocol number AN185688). Briefly, after euthanasia of the pregnant Sprague-Dawley rat ($CO_2$ and bilateral thoracotomy), the brains of embryonic day 18 rats of both sexes were extracted from the skull. The striatum, including the caudate-putamen and nucleus accumbens, were identified (*Banker and Goslin, 1999*). The structures were dissected in ice cold Hank's buffered saline solution Calcium/magnesium/phenol red free (Gibco). Striatum were dissociated in 0.05% trypsin/EDTA (Gibco) for 15 min at 37 °C before 2 washes in Dulbecco's modified Eagle's medium (DMEM, Gibco) supplemented with 10% fetal bovine serum (University of California, San Francisco, Cell Culture Facility) and 30 mM HEPES (Gibco). Neurons were mechanically separated with a flame-polished Pasteur pipette. Nucleofected striatal neurons were transfected using manufacturer's instructions (Rat Neuron Nucleofector Kit, Lonza) for rat hippocampal neurons before plating. Cells were plated on poly-D-lysine coated 35 mm glass bottom dishes (Matek) in DMEM supplemented with 10% fetal bovine serum. Medium was exchanged 1–4 days after plating for phenol red free Neurobasal (Gibco) supplemented with Glutamax 1 x (Gibco) and B27 1 x (Thermo Fisher). Half of the culture medium was exchanged every week with fresh, equilibrated medium. Cytosine arabinosine 2 mM (Sigma-Aldrich) was added at 8 days in vitro (DIV). For transfection using lipofectamine 2000

(Thermo Fisher), transfection was performed on DIV 8, using 1 ml of lipofectamine and 1 µg DNA in 1 ml of medium per 35 mm imaging dish, and medium was exchanged 6 hr later. Neurons were maintained in a humidified incubator with 5% $CO_2$ at 37 °C and imaged after 13–21 days in vitro. All experiments were performed on at least 2 independent cultures.

## cDNA constructs

To generate SSF-STANT in PCAGGS-SE, SSF-STANT sequence was amplified by PCR and inserted into pCAGGS-SE after digestion and ligation using EcoRI and XhoI sites. To generate SEP-DOR, SEP and DOR sequences were amplified by PCR and inserted into PCAGGS-SE using EcoRI and XhoI sites with a two fragments In-Fusion (Takara Bio) strategy. SSF-DOR S/T to A in PCAGGS-SE was generated with two fragments In-Fusion cloning in the NheI site. First fragment was a PCR amplification of the DOR sequence, and the second a codon optimized gene block that encodes the C-terminus tail of DOR with S and T amino acids sequences mutated to encode for A. SEP-DOR S/T to A was generated by PCR amplification of SEP and DOR S/T to A sequences and insertion into PCAGGS-SE using NheI sites with a two fragments In-Fusion strategy. All constructs were verified using sequencing, sequences for primers and gene block can be found in the extended methods section.

## Widefield imaging

Imaging of VAMP2-SEP for measurement of presynaptic activity, as well as imaging of SEP-DOR S/T to A for endocytosis was performed with a S Fluor 40x1.30 NA objective on a Nikon TE-2000 inverted microscope. System was controlled by Micromanager 1.4.10 software equipped with an Andor iXon EM +EMCCD camera, and a Bioptechs objective warmer. Illumination, perfusion and electrical field stimulation devices were custom built (see extended methods and files on the repository https://doi.org/10.5281/zenodo.6954811). Briefly, an insert was 3D printed and equipped with two platinum wires distant of 1 cm used for electrical field stimulation (100 action potential at 10 Hz, 10 V/cm). Illumination in the green channel (VAMP2-SEP, SEP-DOR S/T to A) was controlled and synchronized with an Arduino uno, with a blue LED replacing the mercury bulb of a Nikon lamp and an appropriate combination of filters and dichroic mirror. Illumination in the red channel was achieved with a 543 nm HeNe laser (Spectra Physics) and a micrometer-guided illuminator (Nikon), and syp-mCh fluorescence imaged with an appropriate set of filters and dichroic mirror. Neurons in glass bottom culture dishes were transferred from the incubator onto the system, culture medium removed and exchanged for HEPES buffered saline solution (HBS) imaging solution. HBS contained, in mM: NaCL 120, KCl 2, CaCl2 2, MgCl2 2, Glucose 5, HEPES 10 and osmolarity was adjusted to 270mOsm and pH to 7.4. This insert left a dead volume of 300 µl inside the imaging dish and was used to perfuse solutions with a debit of 1.5 ml/min.

To measure presynaptic inhibition neurons were nucleofected with VAMP2-SEP and opioid receptors. 2 min acquisitions at 1 Hz were performed sequentially and electrical field stimulation starting at frame 59. First acquisition was always in HBS only to obtain a baseline response, ammonium chloride solution (HBS containing 50 mM $NH_4Cl$ with NaCl adjusted to 80 mM) was added with a pipette for the last 20 frames of the last acquisition. To monitor stability of the recording in SSF-MOR expressing neurons ('control', *Figure 1C*), the second acquisition was started 1 min after the end of the first one and was in HBS only. To monitor acute inhibition at MOR, solution was shifted to HBS +DAMGO (Sigma-Aldrich) 10 µM after the end of the first acquisition and the second acquisition was started 1 min later. To monitor tolerance, neurons were subject to the same protocol except that they were pretreated with DAMGO 10 µM directly in the cell culture medium for 18 hr in the incubator before imaging. Because our perfusion system allows for only 3 different solutions, we generated the concentration-response curves using only two concentrations of DAMGO per acquisition. After the baseline acquisition in control solution, a first concentration of DAMGO was perfused for 1 min and through the second acquisition. Solution was shifted to a higher concentration of DAMGO for 1 min before starting the third acquisition, and ammonium chloride solution added for the last 20 frames. Using this protocol, acute inhibition or inhibition after 18 hr of incubation with DAMGO 10 µM was measured in the presence of, sequentially, 0.1–1 nM, 10–100 nM, or 30–300 nM DAMGO. When assessing desensitization after tolerance, the protocol was the same except that neurons were kept in DAMGO for 8 more minutes at the end of the second acquisition, and a third acquisition was performed still in the presence of DAMGO, with ammonium chloride solution added at the end of the

last acquisition. For β-CNA experiments the protocol was the same as for the measure of desensitization after induction of tolerance except that neurons were treated with β-CNA 50 nM for 5 min and washed three times with HBS before imaging, not incubated with DAMGO. To obtain time course inhibition neurons were imaged as described when measuring tolerance except incubation time changed, and for the β-CNA conditions neurons were incubated with β-CNA 50 nM for 5 min and washed three times with media before incubation with DAMGO 10 μM for the time indicated. Acute inhibition and tolerance measurement for neurons nucleofected with VAMP2-SEP and SSF-MOR S/T to A and SSF-STANT was assessed as described for SSF-MOR. For measure of tolerance in the presence of Cmpd101 neurons were incubated with Cmpd101 30 μM (HelloBio) or DMSO control in the culture medium in the incubator for 10 min before adding DAMGO 10 μM for 18 hr and imaging as described previously. Tolerance to morphine 10 μM (Sigma-Aldrich), PZM21 10 μM and TRV130 10 μM (generous gifts from Aashish Manglik, UCSF) with either Cmpd101 30 μM or DMSO vehicle was assessed as described with DAMGO. Measure of acute inhibition, acute desensitization, tolerance, Cmpd101 effect for SSF-DOR and SSF-DOR S/T to A was assessed as described for SSF-MOR except the agonist used was DADLE 10 μM (Sigma-Aldrich). To monitor the lack of cross tolerance between SSF-DOR and SSF-MOR, both receptors were nucleofected together with VAMP2-SEP and DAMGO 10 μM or DPDPE 10 μM (Sigma-Aldrich) added to the culture medium for 18 hr in the incubator before imaging. Neurons incubated with DAMGO were imaged in HBS, solution was changed to HBS +DPDPE 10 μM and a second acquisition started one minute later, last acquisition was started 1 min after switching the solution to HBS +DAMGO 10 μM. Same was done for neurons incubated with DPDPE except the order of solution exchange was switched.

To monitor SEP-DOR S/T to A endocytosis, neurons were nucleofected with syp-mCh and SEP-DOR S/T to A and imaged in HBS for one frame in the syp-mCh channel and one frame in the SEP channel. Perfusion was switched to ammonium chloride solution and a second image taken 1 min later. Perfusion was switched to HBS +DADLE 10 μM and a third image taken 30 min later, the fourth frame was taken after 1 min of perfusion with ammonium chloride solution.

## Oblique illumination imaging

Imaging of surface labeled opioid receptors at retromer marked endosomes, insertion events as well as quantification of surface fluorescence of SEP-tagged opioid receptors in axons was performed on a Nikon Ti-E TIRF microscope controlled by NIS-Elements 4.1 software. Microscope was equipped with an Andor iXon DU897 EMCCD camera, a perfect focus system and an objective and stage heater set to 37 °C. Oblique illumination was achieved with 488, 561, and 647 nm solid-state lasers (Keysight Technologies) coming at an oblique incident angle from an Apo TIRF 100x1.49 NA objective, and all channels imaged with an appropriate set of dichroic mirror and emission filters.

To image the recruitment of surface labeled receptors at endosomes, neurons that were transfected with VPS29-GFP, syp-mCh and SSF-tagged opioid receptors using the lipofectamine method were incubated for 15 min with Alexa-647 conjugated anti-FLAG antibody (1/1000, M1 antibody from Sigma-Aldrich Cat# F-3040, RRID: AB_439712, Alexa Fluor 647 Protein Labeling Kit from Thermo Fisher) before neurons were washed three times with HBS and mounted on the microscope in HBS. One image was acquired in the syp-mCh channel, rest of the acquisition was one frame in the Alexa-647 channel and one frame in the GFP channel every minute for a total length of the time lapse of 35 min. Agonist was added by pipetting 100 μl of agonist-containing solution into the glass bottom dish after 5 min of baseline to a final concentration of 10 μM, as indicated in figure legends. When neurons were treated with Cmpd101 or DMSO, neurons were incubated with Cmpd101 30 μM or DMSO vehicle together with M1-Alexa647 1/1000 for 15 min. Cells were washed three times with HBS and mounted on the stage with HBS +Cmpd101 30 μM or HBS +DMSO vehicle, rest of the protocol was the same as described previously.

For imaging of insertion events, striatal neurons were transfected using the lipofectamine method with SEP-tagged opioid receptors and syp-mCh. Cells were incubated for 20 min in the culture medium in the incubator with DAMGO 10 μM (for SEP-MOR) or DADLE 10 μM (for SEP-DOR), before three washes with HBS and mounting on the microscope stage in HBS +agonist at 10 μM concentration. Cells in the no agonist conditions were washed in HBS and mounted in HBS without agonist. One frame was acquired in the red channel, and the green channel was imaged at 10 Hz in stream mode for 5 min.

For imaging of surface fluorescence from SEP-tagged opioid receptors, neurons nucleofected with SEP-tagged opioid receptor and syp-mCh were washed three times with HBS and mounted onto the microscope stage in HBS. 30–40 random regions of interest were selected per dish in the syp-mCh channel only (experimenter was blinded to the green channel), and one image acquired in the green and red channel for each region. When neurons were chronically treated, cells were incubated for 18 hr with agonist at a concentration of 10 µM in the culture medium in the incubator, the rest of the protocol was the same. If Cmpd101 or DMSO vehicle were present, Cmpd101 30 µM or DMSO vehicle were added to the culture medium for 10 min before agonist 10 µM was added for 18 hr, rest of the protocol was the same. Experiments were performed on the same culture in parallel for the conditions that are presented on the same graphs.

## Image analysis

Image analysis was performed on unprocessed 16 bits TIFF images using custom written scripts in MATLAB (Mathworks, R2019b). Scripts are provided on an open repository at the following address: https://doi.org/10.5281/zenodo.6954811. See extended methods for details of the procedures.

For the quantification of presynaptic activity, putative synapses were detected using an automated image classifier on an average baseline (defined as 6 frames before stimulation) subtracted ammonium chloride average image (defined as 5 last frames of the last acquisition). Fluorescence was quantified in a 5 pixels radius circle around each synapse, baseline fluorescence for each acquisition subtracted and normalized to the fluorescence in ammonium chloride. Synapses were validated if no pixel was saturated and the amplitude of the response during the first acquisition (in HBS only) was five times greater than the standard deviation of the baseline. For each condition and each acquisition, fluorescence from validated synapses were averaged across conditions and displayed in the inset curves (± standard error of the mean). To calculate the degree of inhibition, only acquisitions that had more than 50 validated synapses were considered and their fluorescence averaged across each acquisition. The ratio of the amplitude (defined as the maximum value of the 5 frames after stimulation) over the amplitude of the first acquisition was used to define the degree of inhibition. Normalized concentration-response curves were generated by calculating the mean inhibition for each data point, subtracting the average inhibition observed in control solution (*Figure 1C*) and normalizing by the average acute inhibition in 10 µM DAMGO.

For the quantification of SEP tagged opioid receptors and syp-mCh fluorescence at synapses, synapses in focus were identified based on syp-mCh signal and manually picked, only exceptions were synapses that were over glial background autofluorescence or overlapping with somato-dendritic SEP signal. For each synapse, background subtracted average fluorescence within a 3 pixels radius was calculated for both channels. Values for each synapse and channel were normalized by the median fluorescence of the 'no-agonist' condition of the same experimental day and data pooled between experiments.

To quantify receptor recruitment at retromer marked endosomes, a mask of VPS29-GFP endosomes was generated for each frame of the time lapse. To do so, regions of interest were manually selected on the image and refined by thresholding a maximal temporal projection of the receptor channel. Endosomes within this region were defined by a VPS29-GFP fluorescence value above a threshold set manually and objects larger than 5 pixels. For each region, background subtracted receptor fluorescence was calculated for each segmented endosomes for all frames. Average receptor fluorescence was calculated at all segmented endosomes of the same region, for the 'no agonist', baseline bin value. Receptor fluorescence at each endosome was normalized by this baseline value and averaged across all regions for 5 min intervals as indicated in the figures. Binned values were averaged across acquisitions for the same condition.

To quantify the frequency and fluorescence of single insertion events, bursts of fluorescence were manually selected on the image series. Fluorescence was quantified in a 2.2 pixels radius circle and the average baseline fluorescence of the 10 preceding frames was subtracted. Amplitude of fluorescence events is defined as the maximal fluorescence in the 10 frames following the detection. Fluorescence curves were averaged for all events of the same conditions. Frequency of events were normalized for each acquisition by the number of syp-mCh marked synapses in the field of view.

To quantify SEP-DOR S/T to A endocytosis using ammonium chloride unquenching, the four images were manually aligned to the first image to compensate for drift over the acquisition and lines drawn

on axons identified based on syp-mCh signal. Background subtracted average fluorescence from a 3-pixel wide linescan was calculated for all images and normalized to the fluorescence value of the first image for each acquisition.

## Data presentation and statistics

Quantifications of data are presented as either mean ± standard error of the mean, cumulative frequency curves, paired measurements or box and whisker plots (4 quartiles with inclusive median, outliers are not displayed, average is marked as a 'X'). Graphs were generated with Excel (Microsoft office, 2016). Look up tables used can be found in the extended method section. All experiments were performed on at least independent neuronal cultures and sample size indicated in the figure legends. When performing two tailed Student's t-test the software used was Excel, two samples Kolmogorov-Smirnov test were performed with MATLAB. 95% confidence intervals were estimated by calculation of the mean over 50,000 random bootstraps using MATLAB. Fitting of the concentration-response curves and repeated measure ANOVA were performed using Prism 9 (GraphPad). For receptor enrichment at endosomes, we excluded cells with an enrichment value >3 fold from the statistical analysis because such high enrichment values result from very low baseline and thus introduce excessive variability in the ΔF/F calculation. 3 cells in total were excluded in this manner, for the conditions DOR +agonist (#3318) compared to vehicle (p-value 0.0763 if included), MOR +agonist (#2782) compared to vehicle (p-value 0.0071 if included), DOR +DMSO (#3120) compared to DOR +Cmpd101 (p-value 0.48 if included), and are highlighted in the accompanying figure dataset.

## Acknowledgements

We thank Aashish Manglik for providing aliquots of PZM21 and TRV130. We thank Miriam Stoeber for valuable discussions. We thank members of the von Zastrow lab for valuable discussion, suggestions, and technical help throughout this project. Oblique illumination experiments were performed at the Center for Advanced Light Microscopy at UCSF and we thank its expert staff for providing advice and maintaining the equipment. We thank the Center for Advanced Technology at UCSF for providing access to the 3D printer.

## Additional information

### Funding

| Funder | Grant reference number | Author |
|---|---|---|
| National Institutes of Health | DA012864 | Mark von Zastrow |
| National Institutes of Health | DA010711 | Mark von Zastrow |

The funders had no role in study design, data collection and interpretation, or the decision to submit the work for publication.

### Author contributions

Damien Jullié, Conceptualization, Data curation, Software, Formal analysis, Supervision, Methodology, Writing – original draft, Writing – review and editing; Camila Benitez, Resources; Tracy A Knight, Software; Milos S Simic, Formal analysis; Mark von Zastrow, Conceptualization, Supervision, Funding acquisition, Methodology, Writing – original draft, Project administration, Writing – review and editing

### Author ORCIDs

Damien Jullié ⓘ http://orcid.org/0000-0002-1347-1448
Mark von Zastrow ⓘ http://orcid.org/0000-0003-1375-6926

## Ethics

All procedures were performed according to the National Institutes of Health Guide for Care and Use of Laboratory Animals and approved by the University of California San Francisco Institutional Animal Care and Use Committee (protocol number AN185688).

## Decision letter and Author response

Decision letter https://doi.org/10.7554/eLife.81298.sa1
Author response https://doi.org/10.7554/eLife.81298.sa2

---

# Additional files

## Supplementary files
• MDAR checklist

## Data availability

Supplementary files Figure 1- 6 contain the numerical data used to generate the figures. All codes and sample data are available on the repository https://doi.org/10.5281/zenodo.7226372.

The following dataset was generated:

| Author(s) | Year | Dataset title | Dataset URL | Database and Identifier |
|---|---|---|---|---|
| Jullié D, Benitez C, Knight TA, Simic MS, von Zastrow M | 2022 | Supplementary data, methods and analysis scripts from: Endocytic trafficking determines cellular tolerance of presynaptic opioid signaling | https://doi.org/10.5281/zenodo.7226372 | Zenodo, 10.5281/zenodo.7226372 |

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

## Appendix 1

### Supplementary Information and Extended Methods

cDNA constructs

Primers for SSF-STANT in PCAGGS-SE:

SSF-STANT forward: GCGCCTCGAGATGAAGACGATCATCGCCCTGAGC

SSF-STANT reverse: GCGCGAATTCTTAGGGCAATGGAGCAGTTTCTGC

Primers for SEP-DOR in PCAGGS-SE using In-Fusion two fragments:

SEP forward: ATATCGGTACCTCGAGATGGACAGCAAAGGTTCG

SEP reverse: CCAGCTCCATGGTTTTGTATAGTTCATCCATGCC

DOR forward: AAACCATGGAGCTGGTGCCCTCTGCCCG

DOR reverse: CCTGAGGAGTGAATTCCTCTAGATTATCAGGCGGCAGCGCCACCGCCC

Primers and gene block for SSF-DOR S/T to A in PCAGGS-SE using In-Fusion two fragments:

SSF-DOR forward: ATATCGGTACCTCGAGCTAGATGAAGACGATCATCGCCCTG

SSF-DOR reverse: CGACAGAGCTGGCGGAAGCAGCGCTTGAAGTTCTCGTCCA

Gene block: TGCTTCCGCCAGCTCTGTCGCGCCCCATGTGGGCGACAAGAGCCTGGTGCTC TCAGACGACCCAGACAAGCAGCTGCTCGAGAACGCGTAGCAGCATGCGCACCAGCCGACG GGCCAGGGGGTGGGGCTGCCGCATAACTAGCGGCCGCATGCGAATT

Primers for SEP-DOR S/T to A:

SEP forward: CGATATCGGTACCTCGAGCTAGATGGACAGCAAAG

SEP reverse: GGCACCAGCTCCATGGTTTTGTA

DOR S/T to A forward: ACAAAACCATGGAGCTGGTGCCC

DOR S/T to A reverse: AATTCGCATGCGGCCGCTAGTTATGCGGCAGCCCCACCCCC

Images look up tables

see *Appendix 1—figure 1*.

**Appendix 1—figure 1.** Images look up tables.

### Custom built hardware and acquisition protocol

Insert for perfusion and stimulation

A custom made insert for matTek 35 mm glass bottom dishes with 14 mm coverslips (P35G-1.5–14 C) was designed and 3D printed on a uPrint plus (Stratasys) from AMS plastic, the.stl file necessary for this part is provided. We recommend the use of black material for the insert to minimize artifacts of fluorescence (*Appendix 1—figure 2A*). A rubber O-ring (15 mm internal diameter, 1 mm thickness) is installed at the bottom of the insert, these are standard and can be found at any hardware store (*Appendix 1—figure 2B*). After 3D printing of the part and extensive washing with water (typically a day in >500 ml water with multiple replacement of the water, to wash out any residual plastic and alkaline solution), stimulation wires were set up on the insert. We recovered platinum wires (2x1.5 cm) from a broken electrophoresis device (*Appendix 1—figure 2C*) and soldered to about 5 cm standard electrical hookup wire (2 per insert, *Appendix 1—figure 2D and E*). The wire is inserted into a cut 10 ul pipet tip such that the platinum wires protrude from the tip opening while

the solder and attached hookup wire stay inside. We then cut another 10 ul pipet tip and inserted it in the one containing the electrical wire, to squeeze it and secure the wire in place (*Appendix 1—figure 2F and G*). The platinum wire is bent 90 degrees from the cut pipet tip and inserted into the appropriate semi-open cylinder, with repeat for other electrode (*Appendix 1—figure 2H1*). Using small tweezers, shape the platinum wire so it follows the bottom of the insert and secure the end into the appropriate hole in the insert (*Appendix 1—figure 2K and L*). The stimulation wires are separated by 1 cm once set up. We printed a dish holder using the provided.stl file that fits the stage of the microscope and holds the 35 mm dishes while secured by a rubber band (run it over the middle of the opening, *Appendix 1—figure 2L*).

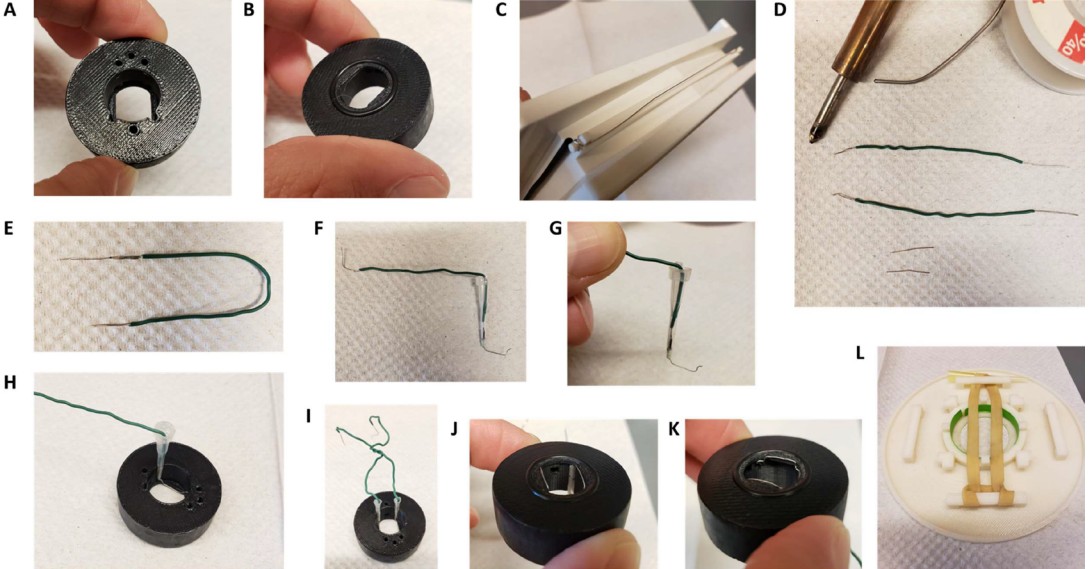

**Appendix 1—figure 2.** Insert for perfusion coupled to electrical stimulation. (**A**) Top view of the insert, with 3 holes for perfusion entry in the insert (top), one hole for vacuum suction (bottom), and 2 semi-open holes for electrodes (bottom). (**B**) Bottom view of the insert with O-ring in place. (**C**) Broken gel electrophoresis with platinum wire. (**D**) Electrodes are made by soldering 2 platinum wires (bottom) to regular electric wire. (**E**) Platinum wire soldered to the electric wire. (**F**) Electrode secured and ready to be installed. (**G**) Close up view of the electrode, note the angle. (**H**) Insert with one electrode installed. (**I**) Insert with both electrodes installed, we recommend you twist the electrical wires together to solidify the installation and make sure the hole for vacuum stays easily accessible. (**J,K**) Bottom view of the insert with the two electrodes installed. (**L**) Dish holder for 35 mm dishes with rubber band installed.

## Zapomatic

This device uses an Arduino circuit board as frame counter and pulse generator, and it steps up the pulse generator output for electrical field stimulation through an optoisolator circuit (*Appendix 1—figure 1–3*). The pinout shown in *Appendix 1—figure 3* is for an Arduino Mega2560 but any Arduino board should work. We use a 30 V regulated 'wall wart' DC power supply to drive the stimulator output and adjust output manually using a 1 kΩ potentiometer as a voltage divider; any buffered constant voltage source that supplies ~30 V at up to ~50 mA should do. The electrical wires of the insert are connected to the zapomatic output using a BNC cable terminated with alligator test clips (see Appendix 1-figure 5G). The voltage and current levels used here are not dangerous but caution is advised to avoid damage in the event of a wiring fault. Individuals lacking skill in electrical construction may want to consult an experienced colleague; the end user is solely responsible for assuring safety to individuals and equipment, and for addressing any issues regarding local electrical code compliance.

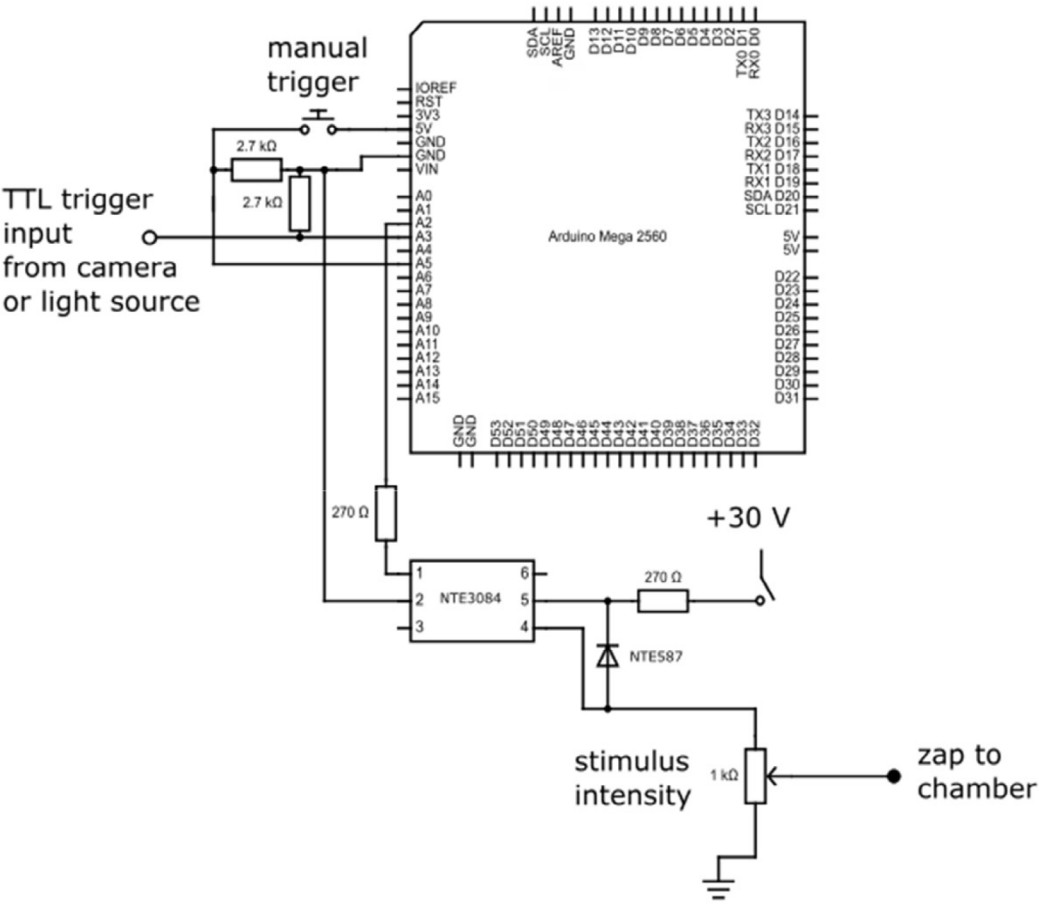

**Appendix 1—figure 3.** circuit diagram of the zapomatic.

Parts detail:
Arduino Mega 2560. Way more than needed for this project (had it handy), any Arduino board should work.
NTE3084. Darlington optoisolator.
NTE587. Fast switching diode.

The provided Arduino code "Dam100AP" is uploaded as firmware to the Arduino. The board receives a TTL high signal from the camera trigger output and counts the number of frames. Once a user-defined frame number is reached (here 59), the zapomatic fires electrical pulses through the BNC connect at a user-defined pulse/frequency/duration (here 100x1ms pulses at 10 Hz). The intensity of electrical stimulation is adjusted using the potentiometer, measured with an oscilloscope connected in parallel from the stimulator BNC output to achieve a stimulus field intensity of 10–12 V/cm. The zapomatic counter resets after the firing sequence, and the zapomatic will fire again any time it reaches the count threshold (just once with our acquisition setting). A manual button on the device can be pushed to trigger firing manually. A LED indicates when the device is firing (same frequency/duration).

Care may be required to appropriately ground the stimulator circuit, depending on your stage setup or associated equipment; the opto-isolated output stage is amenable to flexible grounding. Placing an LED in series with the stimulator output can be convenient. If the LED appears dim when firing, this suggests insufficient current flow in the circuit. Poor current flow can occur due to a poor mechanical connection at the electrodes, such as failure at a solder joint or oxidation on the electrode; check connections and interfaces for proper conductivity.

Frame count is programmed to reset to 0 after 10 s in the absence of a TTL camera pulse.

If counting frames directly from the camera's TTL output, for consistency we recommend disconnecting the zapomatic input or turning off its firing switch until ready to start acquisition.

## Setting up the insert

The perfusion is based on gravity flow from up to 3 different solution sources, each line consisting of an assembly of (from top to bottom): syringe (containing the solution), a 0.45 µm filter, a manual valve, tubing, and ending with a 10 ul pipet tip (*Appendix 1—figure 4*). A separate vacuum line to remove solution need to be setup and attached to tubing ending with a 10 ul tip. We used two 30 ml syringes filled up with ~35 ml of solution. One contained control imaging solution (HEPES buffered saline solution (HBS) adjusted to pH 7.4 containing, in mM: NaCL 120, KCl 2, $CaCl_2$ 2, $MgCl_2$ 2, Glucose 5, HEPES 10 and osmolarity was adjusted to 270 mOsm), the other HBS +drug (typically DAMGO or DADLE at 10 µM). Syringes are placed vertically so the filter is at about the height of the stage (*Appendix 1—figure 4H*). When setting up, use the syringe piston to suction the air trapped in the filter (valve closed), load the whole line with solution, and remove any air bubble. When opening the manual valve solution flows from the end of the line by gravity, adjust the differential in heights between the top of the solution in the syringe and the insert to adjust the flow. In this setting the perfusion runs at about 1.5 ml per minute and we readjust the volume between acquisitions. See *Appendix 1—figure 4* for a schematic of the whole setup.

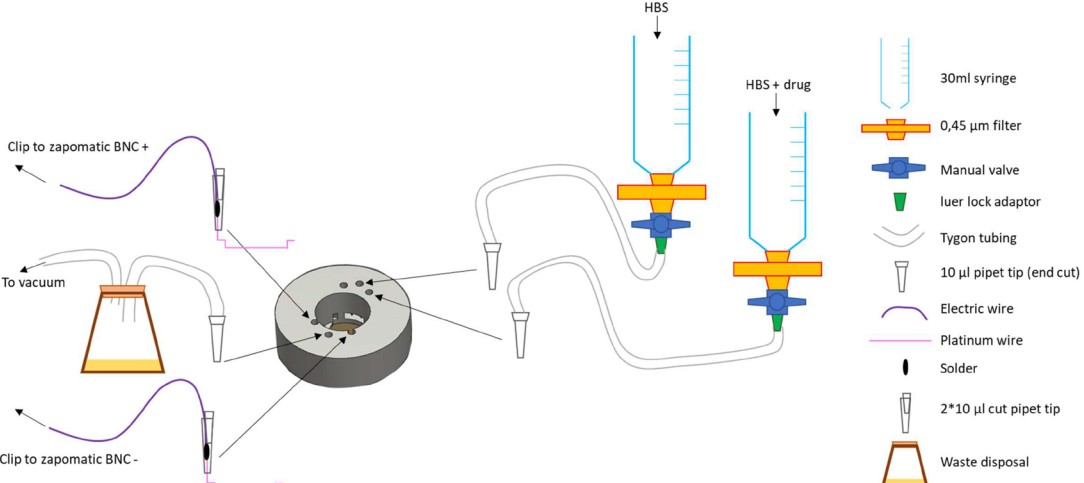

**Appendix 1—figure 4.** schematic of the perfusion installation.

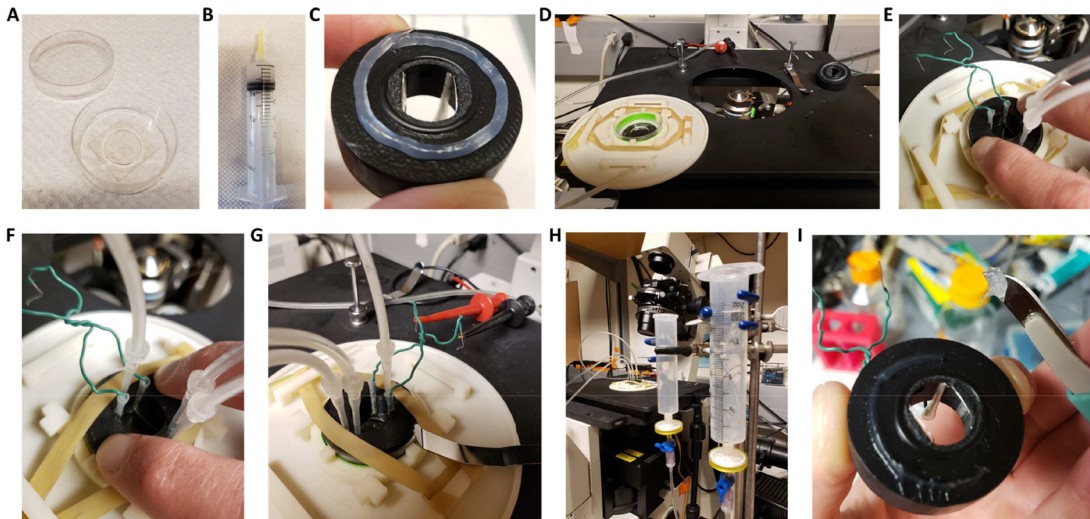

**Appendix 1—figure 5.** Setting up of the insert. (**A**) 35 mm glass bottom dish with 14 mm coverslip, note the plastic area surrounding the coverslip that needs to be completely dry for the vacuum grease to create a good seal. (**B**) Vacuum grease loaded into a 30 ml syringe completed with a cut pipet tip. (**C**) Grease applied to the bottom of the insert. (**D**) Dish setup in the holder with the insert ready to be installed. (**E**) Insert with perfusion lines installed, keep pressure on the insert to maintain a good seal. (**F**) Vacuum line installed and rubber band securing the insert, after this step it is no longer necessary to keep pressure on the insert. (**G**) Dish older with insert installed on the optics and electrodes connected to the alligator test clips. (**H**) View of the whole system in place. (**I**) After removing the insert use a scalped to remove the excess grease for the bottom of the insert, finish cleaning with a kimwipe.

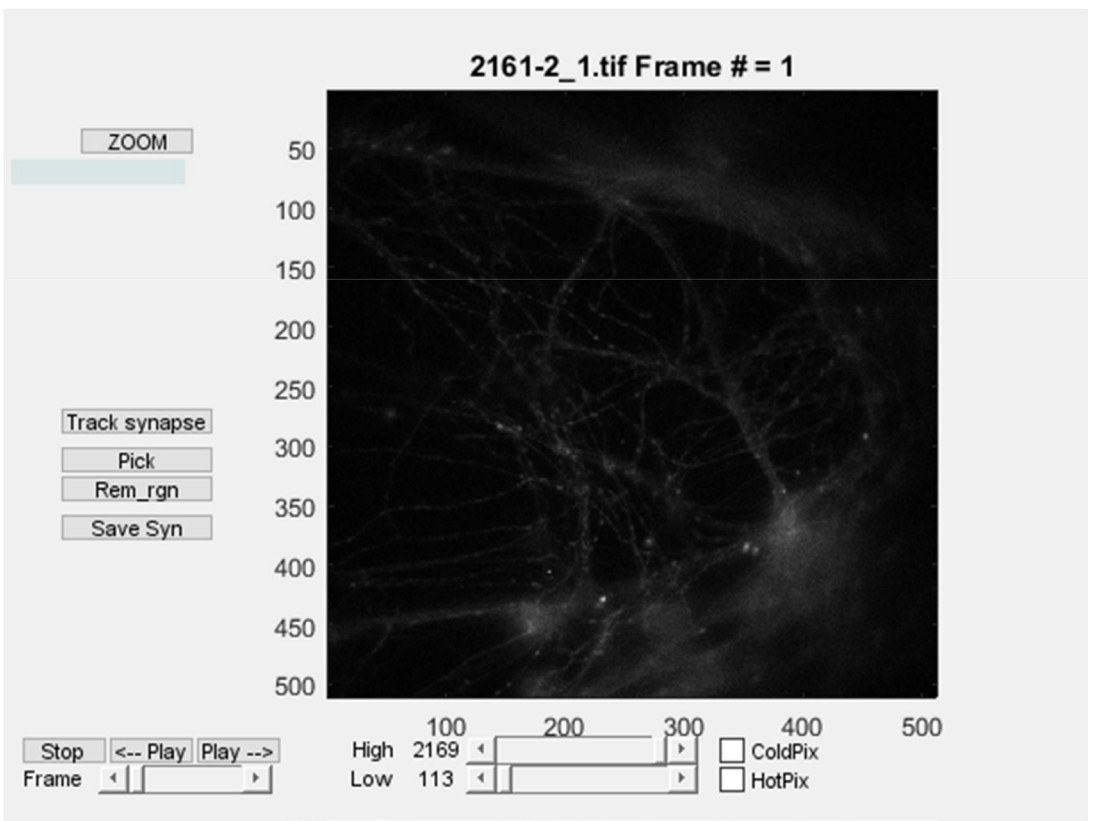

**Appendix 1—figure 6.** Zap100_1_2 user interface. Title is name of the file, # indicates the frame (here 1). Lower list of buttons (Track synapses, Pick, Rem_rgn, Save Syn) are controls to select synapses. Lower left buttons are controls for the frame and play the movie. Middle lower controls are to set the contrast.

If the vacuum does not work efficiently there is a risk of overflow and microscope damage.

The mounting of the dish is done on the microscope stage but away from the optics.

Primary neurons were cultured in matTek 35 mm glass bottom dishes with 14 mm coverslips (P35G-1.5–14 C, *Appendix 1—figure 5*). Insert is mounted into Using Molykote vacuum grease (Motion industries, # 00785543) inside a syringe (*Appendix 1—figure 5B*), create a uniform ~2 mm thick layer of grease surrounding the O-ring (*Appendix 1—figure 5C*). The dish is brought to the microscope from incubator (37 °C 5% CO2) and media inside the dish removed as extensively as possible while not approaching the suction close to the glass part to prevent drying of the live cells (*Appendix 1—figure 5D*). The plastic part of the dish is further dried using a rolled kimwipe (important to make a good seal), and the insert pressed onto the dish. Do not release the pressure until the end of installation. Setup the perfusion lines in the appropriate entries and open the valve for the vehicle solution (*Appendix 1—figure 5E*). Wait for it to flow above the platinum wires and place the vacuum line in the appropriate hole. The insert is designed for strong suction such as a vacuum pump connected to a waste container. Suction should be steady and volume inside the insert constant, weak suction can result in changes in volume of solution inside the insert which cause focus issues. This insert left a dead volume of 300 ml inside the imaging dish. Once the perfusion and vacuum lines are setup, place the rubber to secure the dish and place the dish holder on the stage (*Appendix 1—figure 5F*), and use the alligator test clips to connect the electrodes to the zapomatic (*Appendix 1—figure 5G*). If using a different dish holder, we recommend you clamp the insert on the dish to prevent movement. The acquisition is ready to start, make sure the solution lines are never empty. At the end of the acquisition, remove the solution and vacuum lines from the insert, place the ruber bands on the blocks away from the insert, and pop the dish out of the holder by pressing the dish holder on something into the objective opening. Gently take the insert out of the dish being extra careful to not displace the electrodes and use scalpel or any flat blade to remove the grease from the insert (*Appendix 1—figure 5I*), finish cleaning the bottom of the insert with a kimwipe. After each day of acquisitions, the perfusion lines are abundantly rinsed with distilled water (typically >150 ml per line) with the insert and vacuum line in place in the last imaged dish. Using the plunger all lines are emptied from remaining water (pump air into the filter/tubing, remove syringe from the line, load air into the syringe, couple to line, push the plunger. Repeat a few times).

## Microscope and acquisition software

Imaging of presynaptic activity in striatal neurons that were nucleofected opioid receptors together with VAMP2-SEP was performed on a Nikon TE-2000 inverted microscope, objective was a S Fluor 40x1.30 NA objective, an Andor iXon EM +EMCCD camera and a Bioptechs objective warmer. Widefield illumination in the SEP channel was controlled and synchronized with an Arduino uno, with a blue LED replacing the mercury bulb of a Nikon lamp and an appropriate combination of filters and dichroic mirror. Acquisition software used was micro manager Version 1.4.10.

## Acquisition settings

For most experiments described in this study, the acquisition protocol was standard and consisted in two acquisitions of 120 frames a 1 Hz (second acquisition usually started 1 min after switching the valves from vehicle solution to vehicle +drug). Camera was encoding images at 14 bits and saved as 16 bits 512*512.tif images. The Arduino code embedded in the zapomatic triggered 10 Hz – 1ms stimulation starting at frame 59 for a total of 100 stimulations – 10 s. Control imaging solution was then switched to imaging solution +drug by switching valves on syringe assemblies (open first and close second to ensure continuous flow of solution) for one minute before starting the second acquisition. At the end of the second acquisition, 1 ml of a solution containing ammonium chloride (HBS containing 50 mM NH4Cl with NaCl adjusted to 80 mM) was pipetted in the open chamber after closure of the perfusion valve, starting at frame 100.

Time in presence of agonist or sequence of HBS +drugs varied across experimental designs as described in the manuscript. When a third acquisition was performed NH4Cl solution was pipetted at the end of the third acquisition.

## Analysis of presynaptic activity

All the workflow presented here is setup for analysis in batch (software handles multiple acquisitions at the same time). User will need to install the different scripts provided and add them to the path

in MATLAB. In addition, it requires the image processing toolbox, the machine learning toolbox and the parallel processing toolbox (optional but slower, see notes). Sample data are available for running the analysis.

It is likely that some of the scripts will not function on mac computers. One reason is indexing of folders that uses " \ " instead of " / ", user can edit the code manually and exchange the symbols to solve these issues. Another reason is that the "xlswrite" function that is not compatible with the OS, to solve this convert the cell array to table and use "writetable" function instead. An example of the syntax is provided below:

```
sheet                          =                          array2table(RESULT);
writetable(sheet,['myfile.xlsx'],'Sheet',['sheetName'],'WriteVariableNames',0);
```

## Detection of synapses

### Manual mode

Set the current folder containing the last movie of the acquisition with NH4Cl at the end ("Acquisition#" in the sample dataset). In the command window, enter "**zap100_1_2**". Program will ask to select the corresponding file (.tif file ending in –2_1.tif in the sample dataset) and a graphical user interface will appear (*Appendix 1—figure 6*).

The manual workflow is as follows. Review the movie for quality first. When clicking on "track synapses", select "yes" to manually pick, and program will ask to identify a background region (rectangular), and will automatically go to frame 121, the last one. This frame does not belong to the original movie but instead is a differential image of the last 5 frames of the movie (in NH4) minus an average of the 10 frames that precede the stimulation +8,000 (AU). These parameters can be found 27–31 of the code. Synapses are much easier to identify on this image. Any left click on the image will select a synapse, that appears as a green circle. Right click will stop this sequence. You can select more synapse anytime (after zooming if needed) or remove synapse one at a time using the "pick" button. Right click when "pick" button is activated removes synapses. Use Rem_rgn to remove multiple synapses within a rectangular region. When satisfied with the selection, save the list of synapses using "Save Syn". It will generate a.txt file that ends with ZAP1_2_syn.txt that contains the coordinates of synapses as follow:

| | |
|---|---|
| 59 5 10 5 | parameters for the diff NH4 image |
| 69 325 46 15 | coordinates of background region |
| 1 120.78 160.6 0 | synapse 1 X Y coordinates |
| 2 163.04 171.98 0 | synapse 2 X Y coordinates |
| 3 200.42 165.48 0 | synapse 3 X Y coordinates |
| And so on… | |

## Using machine learning

### Step1: generate sample synapse images

For generating the sample images it is important that you select ALL synapse-looking structure on the image when selecting them manually. Anything on the image that isn't registered as a synapse will be treated as background by this code. Organize your current folder as follow (as organized in the sample dataset, standard for all batch processing), that usually reflect multiple acquisitions (folder 1.1,1.2 etc.) for multiple conditions (folder 1, folder2):

Code as setup will recognize the movie with NH4Cl at the end because the name finishes by 2_ 1.tif (line 46). This is the way micromanager saves the second acquisition in our system. Either adapt code to a unique identifier for NH4 movie or add "2_1.tif " to the file name.

| | |
|---|---|
| Folder 1 | CONDITION FOLDER 1 |
| Folder1.1 | ACQUISITION FOLDER 1.1 |
| .tif movie of first acquisition (acquisition 1) | |
| .tif movie of second acquisition with NH4 at the end (acquisition 1) | |
| .txt file with annotated synapses (acquisition 1) | |
| Folder1.2 | ACQUISITION FOLDER 2.2 |
| .tif movie of first acquisition (acquisition 2) | |
| .tif movie of second acquisition with NH4 at the end (acquisition 2) | |
| .txt file with annotated synapses (acquisition 2) | |
| And so on… | |

                    Folder1.3
            ...
        Folder 2                                        CONDITION FOLDER 2
            Folder2.1
        ACQUISITION FOLDER 2.1
                    .tif movie of first acquisition (acquisition X)
                    .tif movie of second acquisition with NH4 at the end (acquisition X)
                    .txt file with annotated synapses (acquisition X)
            Folder2.2                                    ACQUISITION FOLDER 2.2
                .tif movie of first acquisition (acquisition Y)
                    .tif movie of second acquisition with NH4 at the end (acquisition Y)
                    .txt file with annotated synapses (acquisition Y)
        And so on
        Folder 3
        Enter in the command window "**MLsynGen**"

A few parameters are embedded in the code, same ones used for the differential NH4 image (line 12–14) and "boxsize" (defines the size of each generated image, default is 7) and "samplesize" (how many images to generate from each acquisition, default is 50). As it is set up, code will randomly pick 50 synapses per acquisition at most (line 9) and will generate 3 stacks (it can take time depending on the amount of images you have, be patient):

Stack 1: sampleSyn.tif, the multi-tif stack contains synapse images so central pixel is centered around the manually selected coordinates, and a boxsize selection of the default NH4 image around it (so by default, size of the stack is 15*15*n images, n depending on samplesize or min amount of selected synapses, 15 being 2*boxsize +1).

Stack2: sampleEmpty.tif,, the multi-tif stack contains images selected randomly around synapses (this is a "close by" sample picked from a distance of boxsize away from manually picked synapses), size of the stack is 15*15*2 n images.

Stack3: sampleBack.tif, the multi-tif stack contains images selected randomly in the background, size of the stack is 15*15*2 n images.

Provided sample multi-tif stacks used to train the classifier used in this study have n=20,643 images.

## Step 2: train and run the classifier

In the condition folder, enter "**HogClassBatchGaussAl**" in the command window.

Code will ask to select a synapse movie, pick sampleSyn.tif

Then to select "Off movies" 1 and 2, select sampleEmpty.tif and sampleBack.tif (order does not matter).

Code will train a support vector machine classifier that is fed a histogram of gradient vector after transform of the images (enlarge +gaussian filter +Fourier processing) and an intensity value (center – background around). All parameters are found line 21–26. It is trained for 2 classes based on the images provided, synapse (1) and not synapse (0). A confusion matrix is displayed once the classifier is trained, and classifier is automatically saved in the current directory, "HogFFTclass.mat".

Classifier will analyze all provided acquisitions, this mode does not run on parallel processing and is slow. To detect synapses, a sliding window operates with a 1.5*boxsize (parameter divwind, 1.5 is default) step and generates an image that is filtered for signal intensity (uniformly high signal or nearby "negative value" that indicate movement are excluded, line 273) and classified as synapse or not synapse. If classifier found a synapse, it will operate a 2D gaussian fit to center the localization of the synapse. Parameters of the gaussian fit are found on lines 287–289 and will need to be adjusted depending on your acquisition setting (typically create an average projection of your "synapse" sample and extract parameter from this). Script will then remove double detections (<tooclose, 3 pixels is default).

For each movie it will generate a "MLsynHFA.txt" synapse coordinate file in the Folder X.Y of each acquisition. You can visually check the quality of the classifier by running Zap100_1_2 in the command window (select "no" to manually annotate, "yes" to load the background region and select original manual file, or select "no" and manually pick a background region). You can manually

edit the synapses at this stage, and save the file as explained before. If classifier performance looks good you can use it directly, no need to re-train every time. The classifier used in this study is provided in the "sample for classifier" dataset.

All parameters provided with this code have been set through iterations and will vary depending on your imaging system (magnification etc.). Make sure your classifier gives great performance on selected data before generalization of its performance to new datasets (>95% accuracy on both classes as defined from classifier output).

### Step 3: run classifier on new data

Organize the dataset to analyze in the current directory as described previously for CONDITION (multiple folders for each acquisition). Copy the classifier generated from step 2 into the current folder, enter "ClassGaussAllign" (ClassGaussAllign3 if 3 acquisitions) in the command window. This code runs on parallel processing, but you can change "parfor" line 31 for a "for" statement and code will run. Classifier will detect synapses as explained previously and generate a "_CGA.txt" file that contains the synapse coordinates. User will need to select background region using ZAP100_1_2 (select "no" to manually annotate, "no" to load background region, and manually pick a background region). While I could have automated this step, I find it useful (and fast) to select the background region while checking quality of synapse detection and eventually adding – removing synapses manually.

## Quantification of fluorescence and normalization of the data:

Set the current folder as described previously (and as orgamized in the sample dataset), with the multiple conditions folders and acquisitions subfolders (2 movies +1 "ZAP" synapse +background coordinate file). Code looks for "ZAP" in the filename to identify the coordinate file, for "2_1.tif" to identify the second file with NH4 at the end, and pick the other ".tif" file as default for the first control movie. You will to rename the files or change the code accordingly (lines 92–94). Enter "zap100Poolstdev" in the command window ("zap100Pool3stdev" if 3 movies). Code requires "xlswrite" function that (to my knowledge) does not work on MAC OS version of MATLAB. User can convert the corresponding cells to tab format and write excel or csv file. I will expand here considering the 2 acquisitions and a note can be found for 3 acquisitions paradigm at the end. Code needs "numinputdlg" script to run a prompt for parameters, alternatively delete lines 50–59 and input parameters manually in lines 41–47. While a number of parameters are embedded in the code for the sake a more friendly user interface, I will provide line numbers for these.

Code will analyze each acquisition, normalize and pool the data for the whole condition, and normalize again. Details of the operations are found below:

For each acquisition, code will generate a.xlsx file containing multiple tabs

First tab contains the parameters, the defaults are:

- First stimulation: default 59, in our setting, marks the beginning of the stimulation
- Size of the synapse: default 5, radius in pixels around synapse coordinates in which the fluorescence measurement is done
- baseline: default 6, number of frames before stimulation to define fluorescence baseline
- Amplitude: default 5, number of frames to define the amplitude of fluorescence increase after stimulation
- Stimulation: default is 10, length of stimulation in frames.
- Frames in NH4Cl: default 5, to define fluorescence in NH4Cl.
- Threshold Amplitude: default 5, to select only responsive synapses.

How these numbers are used for quantification is explained below.

For each coordinates, the script will calculate raw fluorescence values (average fluorescence in a circle of radius "Size of synapse") and subtract the average fluorescence in the user defined background region. This appears in two tabs with "_raw" added at the end of the movie file name.

The script then normalizes the fluorescence values for each synapse. To do so it defines the baseline fluorescence as the median fluorescence value between the frames "first stimulation – baseline" and "first stimulation". Baseline fluorescence value is subtracted from raw values. Script will calculate the average raw fluorescence of the last 5 frames of the NH4 application, and this value is used to divide the baseline subtracted raw fluorescence values, for the two movies. This appears in two tabs with "_NH4norm" at the end, normalized fluorescence over fluorescence in NH4.

In these tabs there are two extra columns at the very end that are labelled "amplitude" and "STAMP". These values are only used for classifying synapses as responsive. Amplitude is the average of _NH4norm fluorescence between the frames (First stim +Stim) and (First stim +Stim + Amplitude), with the 20% highest and 20% lowest values excluded (line 247) to minimize noise. "STAMP" corresponds to this amplitude value divided by the standard deviation of the baseline _NH4norm fluorescence between the frames "first stimulation – baseline" and "first stimulation".

Based on these calculations, a number of synapses are excluded.

- Synapses that have values above the NH4 average value before NH4 is applied (20 frames, line 215).
- Synapses that contain 1 or more saturated pixel (value >16,300 based on 14 bits, line 215) during the last NH4 frames.

These two categories of synapses will appear in the tab "_removed" of the excel file with a value 1 for "saturation".

-Synapses for which STAMP <Threshold for the first acquisition, this is to remove structures detected as synapses that do not respond to the stimulation (low signal to noise). This will appear in the tab "_removed" of the excel file with a value 1 for "amplitude".

The two remaining tabs are labeled "_final" for each movie and contain the quantifications only for the synapse that passed these exclusion criteria.

For each condition, the script will pool data from this single-acquisition analysis and perform another normalization for the whole condition folder, the results are found in a "_std_Pool.xlsx" file.

All quantifications for synapses that have been validated are pooled into two tabs, one for each movie (1st Stim, 2nd Stim), with the name of the corresponding acquisition file in the first column. Last column represents the amplitude of the response as defined previously. Average curves +/-SEM presented in the manuscript were obtained from this dataset.

For each acquisition/movie, an average of fluorescence and amplitude for validated synapses is calculated, this appears in the tabs "1st cell, 2nd cell", as well as the number of validated synapses for each acquisition "n". Reminder, the average amplitude here is not what was used for final quantification, we choose to quantify the amplitude of the average fluorescence - not the average of the amplitudes, to improve signal to noise. This is explained below.

For each acquisition, we therefore obtain two normalized over NH4 fluorescence curves for validated synapses (1st movie baseline, 2nd movie +drug, usually). We here define the amplitude of the average as the maximum fluorescence value between the frames (First stim +Stim) and (First stim +Stim + Amplitude), that is calculated for each movie. This appears into the "final" tab of the excel file under "Max Amp", for each movie (1st STIM, 2Nd STIM). Average fluorescence curves for each movie are normalized to the Max Amp value of the first (control) acquisition and displayed into this "final" tab, as well as averages and SEM value for the whole condition. To define the degree of inhibition in the present manuscript, we only considered acquisitions with at least 50 validated synapses and used the ratio such that the percentage of inhibition is given by:

100*(1-(Max Amp(2nd movie) / Max Amp (1st movie, control))).

In the 3 movies/acquisition paradigm, we used "**zap100Pool3stdev**", with folder containing 1 baseline movie "1_1.tif", 1 drug movie "1_2.tif", 1 drug movie +NH4 at the end "1_3.tif", 1 synapse. txt coordinate file "ZAP". Script will process as previously for 2 acquisitions except that it quantifies another movie but all synapse validation and normalizations steps remain the same.

## Analysis of receptor recruitment at endosomes

The code runs by entering "**Manon**" in the command window of MATLAB.

If you run the code multiple times on the same movies, make sure that the previous files generated by the program are in a different directory or code will overwrite and edit.

**Code asks for "size of the structure",** this is the minimal size in pixels of the structure you would be segmenting. Default is 5, and this is what was used in the study.

**Code asks for synapse movie**, this should be a multi-tif single channel. Code does no operation on this image; it is for display only. If only two channels have been aquired, provide a movie of the same size (repeat marker or receptor for example). It will appear in blue in the image display.

**Code asks for marker movie**, this should be the movie you try to segment, it will appear in green in the image display.

**Code asks for receptor movie**, this is the signal you try to quantify, it will appear in red in the image display.

This is the user interface for this analysis that will pop up (*Appendix 1—figure 7*).

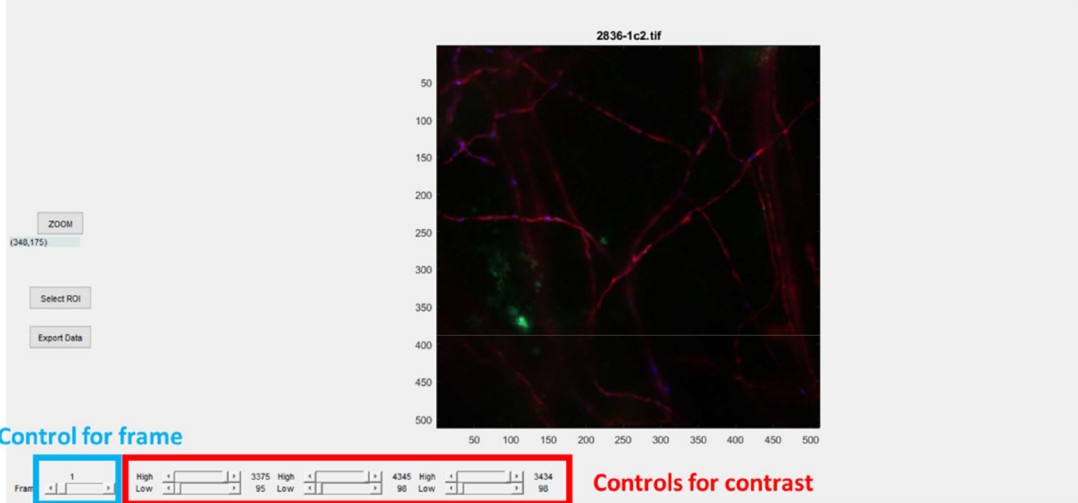

**Appendix 1—figure 7.** User interface for Manon.

If not: make sure all your movies are the same size (frame number in particular), make sure you set the path correctly (for the code and for your working directory), make sure you have the proper toolbox installed (image processing toolbox)

Set contrast and look at your images to check for quality using the controls. Zoom button can be used to zoom on some area of the image, use right click/reset to original view to get back to original image.

Quantification starts by clicking on "select ROI".

Program asks to select a rectangular background region, draw the rectangle with the cursor and double click on the rectangle to validate the background region.

Program returns control of a cursor, use it to draw a polygon around a ROI (left click), double click to close and validate the polygon. Another interface will pop up (*Appendix 1—figure 8*).

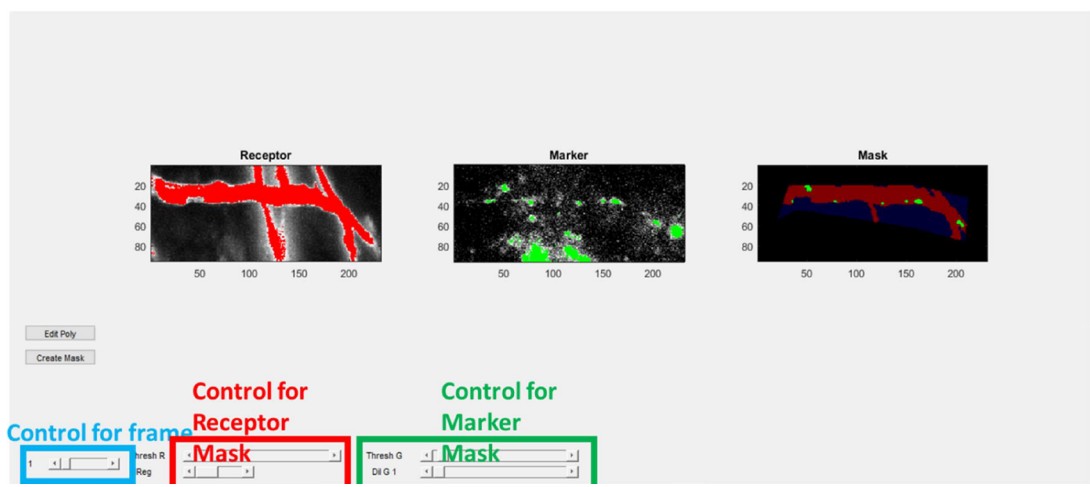

**Appendix 1—figure 8.** segmentation interface for Manon.

Use the controls to define the segmentation

The receptor mask is made on the maximal projection of the receptor movie (red, image on the left). User can control a threshold above which you define the mask, and the number of regions to include in the mask starting from the biggest. This mask appears in the "mask" window as red over

blue background (the polygon ROI). This mask is used to refine the segmentation in the marker channel.

The marker mask is generated using two controls, I recommend you use the threshG control only that sets the threshold for the segmentation. Other parameter for segmentation would be the size of the object (inputed at the beginning of the code, 5 in this study) and dilG that lets you soothe the structures (1 in this study). The segmented structures within the receptor mask (middle, green) appears on the "mask" window as green dots over red background.

When satisfied with the segmentation, click on "create Mask". The window closes and the polygon you drew appears stably on the image. If you're not satisfied with the ROI you selected, click on "Edit Poly", it will erase the polygon like nothing ever happened.

When you have selected all the ROIs on the image, click on export data. Leave some time for MATLAB to do its job as writing on excel takes a bit of time, script will display the message "done", only then you can explore your results in the excel file. The file is added to the current directory and is called "name of your synapse movie_Manon.xlsx". This file contains the quantifications:

In the first tab, every column is a frame of the movie.

**back Fluo** is the average fluo in the background region.

**Average Fluo Receptor** is the background subtracted average fluorescence of the receptor in the max projection mask, per ROI (multiple lines, per ROI).

**Average per region Fluo at marker** is the background subtracted average fluorescence of receptor at endosomes within the same ROI. Values at individual segmented endosomes within that ROI are averaged. (multiple lines, per ROI)

**Average per region norm Fluo at marker** is the background subtracted average fluorescence of receptor at endosomes normalized by the background subtracted average fluorescence of the receptor within the same ROI. Normalized values at individual segmented endosomes within that ROI are averaged. (multiple lines, per ROI)

**Total Av per region Fluo at marker** is the background subtracted average fluorescence of receptor at endosomes within the same ROI. This averaging is done by averaging all pixels for endosomes (compared to averaging individual endosomes) within the ROI.

**Total Av per region Norm Fluo at marker** is the background subtracted average fluorescence of receptor at endosomes normalized by the background subtracted average fluorescence of the receptor within the same ROI. This averaging is done by averaging all pixels for endosomes (compared to averaging individual endosomes) within the ROI.

**Per endosome Fluo at marker** is the background subtracted average fluorescence of receptor at endosomes across all ROIs. SEM is provided.

**Per endosome Norm at marker** is the background subtracted average fluorescence of receptor at endosomes across all ROIs, normalized by the background subtracted average fluorescence of the receptor in the corresponding ROI. SEM is provided.

Fluorescence at individual endosomes is found in sheet 2. First column will be the region ID, first line is average with sem below (if missing endosomes in one region this line gets buggy, sorry I did not fix this…but basically it is the average of all endosome values across all regions and associated sem). **This is the dataset that was used for further analysis in this manuscript with PoolManonBin**.

Normalized Fluorescence at individual endosomes is found in sheet 3. is the background subtracted average fluorescence of receptor at every endosomes across all ROIs, normalized by the background subtracted average fluorescence of the receptor in the corresponding ROI.

Sheet 4 contains all parameters used for the quantification, including thresholds and polygon coordinates.

Program also generates a.tif 8 bit movie "name of your synapse movie_mask.tif" that contains the generated segmentation as follow: polygon value = 1;

Receptor mask value = 2;

Endosome mask value = 3;

In the current manuscript, data were binned by time intervals, normalized and pooled. To do so, Excel files generated by the "Manon" script are then sorted into separate folders per condition. Set the current directory in the directory that contains these folders and enter "**PoolManonBin**" in the command window.

Code goes directly into "sheet 2" of each excel file, for each region it will obtain the average value from all detected structures of the first 6 frames. All values for this region are normalized by this average, and this is repeated for all regions. Normalized values are then averaged per bins frame 0–5, 6–10, 11–15, 16–20, 21–25, 26–30, 31–35 (time lapse were 35 minutes in these experiments). This is repeated across Excel files present in the folder, and code will output a "NameofFolderManon_PoolBin.xlsx" file that contains in the first column the name of the excel file pooled and the binned values for each file, together with an average and sem.

## Analysis of SEP unquenching

The 4 images per acquisition were assembled into a multi.tif stack. In a folder that contains this stack together with the first image, run "**ManuAllign**" in the command window. Code asks for the reference image (first image in our case), then for the stack to align (the 4 images stack). Reference images appear in green on the user interface, red is the stack to align. You will find controls to set the contrast and the frame, as well X and Y adjust that allow you to move each frame of the stack to it is corrected for drift compared to the reference image. When satisfied with the drift correction click on "export" and script will generate a.tif file "originalstackname_alligned".

Code will erase pixels (value set to 0) as you move the image to keep dimensions consistent. These areas were excluded from quantification by staying away from the edges.

To do the linescan analysis on the stack, we used a 4 images stack of the original syp-mCh image (duplicated 4 times) and the aligned stack. Set the current directory in a folder that contains these two files, enter "**Linescan2**" in the command window. Code asks for the stack to quantify first (in green) then for the synapse image (in red) that will appear color coded as indicated on the user interface.

This interface has been my "do everything" interface and contains many buttons, most of which will be buggy when transferred from one code to another, therefore I will not develop here what these buttons do and will focus on the ones that are relevant to this study.

You will find controls to set the contrast and frame as usual for this interface. Click on "axon" button to start manually drawing a linescan using the cross to select points with left click, to stop drawing use the right click for the last point. You might want to draw other lines, just repeat using the "axon" button. Do not change frame, and do not try to interact with the interface when drawing. When done selecting new lines, click on "Quantax" button, code will ask for the width of the linescan (3 in the current study), then select a rectangular background region and double click on it to validate. When done the code will display all kind of boxes around the lines that were visual controls for the linescan operation (no option to set the width of linescan with MATLAB, we had to rotate a rectangular matrix etc.). Script generates an Excel file "line_nameofthefile.xlsx" that contains the average fluorescence value of the linescan for each frame of the stack. Here is how it is calculated:

For each pixel along the line, an average is done along the width of the linescan. Then all linescans values are averaged by the total length of all the different linescans.

## Analysis of SEP surface fluorescence

In a current directory that contains your marker multi-tif file (synapse) and your surface fluorescence multi-tif file of the same number of images (SEP channel), enter "**CircleQuant2**" in the command window. Code asks for the SEP stack first (in green) then for the synapse image (in red) that will appear color coded as indicated on the user interface. To select synapses click on "Pick …", script will ask to select the background region, select synapses with left click, use right click when you are done selecting all synapses on the image. Move on to the next frame using the bottom left slider, repeat operation for each frame by clicking on "Pick …". When the whole stack is reviewed, click on "Quant …", code asks for the diameter of the circle in which to quantify the fluorescence (default is 3, what we used in the current study). An Excel file "nameofgreenchannel_CircleQuant2.xlsx" is generated. Every line of the file displays name of the green stack, frame, synapse number, X coordinate, Y coordinate, average fluorescence in the green channel, average fluorescence in the red channel.

To estimate confidence intervals we used the script "**PermutToxls**". Variables for each measurement (list of fluorescence values) need to be saved beforehand as a.mat file. When running the code, enter the number of iterations N (50,000 in this study) and the name of the two variables you want to compare. The script generates N random bootstraps from each sample distribution and calculate the mean, and outputs the 2.5% (CI low) and 97.5% (CI high) (95% confidence interval values) in a

excel file. The code can also generate random permutation statistics on a N permutations using code retrieved from Laurens R Krol (2022). Permutation Test (https://github.com/lrkrol/permutationTest; *Krol, 2021*), GitHub. Retrieved May 24, 2022 (not used in the present study).

This interface has been our "do everything" interface and contains many buttons, most of which will be buggy when transferred from one code to another. If you need to save the quantification before you are done reviewing the whole stack, you can save the data and pick up where you left by going directly to the frame where you stopped. In this case, make sure to change the name of the first saved.xlsx file or it will be edited and the quantification lost.

## Analysis of SEP fluorescence bursts

Movies are reviewed manually in a directory containing your synapse image and the SEP channel for the apparition of fluorescence bursts in the green channel using the "**CircleQuant2**" script as described above. Using the "Pick" button (lower one), events are selected for the frame where the burst of fluorescence appear. When done reviewing the stack, click on "save" and program will output a "nameofthegreenstack_annotate.txt" file that contains the events coordinate (event ID, frame, X coordinate, Y coordinate). To review the events and quantify the fluorescence, enter "**FT1cTIF**" in the command window. This script is adapted from *Jullié et al., 2014*. Code will ask to input parameters and displays a user interface the allows to browse selected events and displays quantifications. Click "next" to review all events, click "writeXLS" to export the quantification in a "nameofthegreenchannel_data.xls" file. The quantification is to be found in the "green sheet", are displayed the quantification parameters, the integrated fluorescence intensity in a "radius" defined circle around the selected event with the baseline (average fluorescence in the same area for the "baseline" frames before). Area (for average fluorescence) is found in the excel sheet. We used an unreasonably high "threshold parameter" (50) in this study to maintain the size/location of the fluorescence quantification. We obtained the average fluorescence by dividing the integrated intensity by the corresponding area value.

