## [Editor Report]

This manuscript examines the inhibition of transmitter release induced by the activation of opioid receptors, both MOR and DOR, using a novel imaging method. The authors specifically examine how the inhibition of transmitter release is changed following prolonged exposure to saturating concentrations of agonists and they showed convincingly that there is a depletion of plasma membrane-associated receptors and suggest that the decline in receptors at the plasma membrane underlies presynaptic tolerance. This work addresses a long-standing question about how tolerance develops at the presynaptic level and indicates that the location of receptors is critically important in the development of tolerance. This work is fundamental and a game changer in the understanding of tolerance at the cellular level.

---

## [Decision Letter]

**Decision letter after peer review:**

Thank you for submitting your article "Endocytic trafficking determines cellular tolerance of presynaptic opioid signaling" for consideration by *eLife*. Your article has been reviewed by 3 peer reviewers, and the evaluation has been overseen by a Reviewing Editor and Michael Taffe as the Senior Editor. The following individuals involved in the review of your submission have agreed to reveal their identity: John T Williams (Reviewer #1); Matthew Ryan Banghart (Reviewer #2); Susan L Ingram (Reviewer #3).

Essential revisions:

1) Include discussions about the potential impact of overexpression of receptors and membrane protein dynamics.

2) Incorporate the suggestions made by reviewers on the presentation.

*Reviewer #1 (Recommendations for the authors):*

1. The decline in the ability of DAMGO to reduce transmitter release following long-term incubation is clear, convincing, and well presented. It seems that in order to establish a strong conclusion for the development of tolerance the construction of even a rudimentary concentration-response curve following long-term agonist incubation would be valuable. That is long-term incubation with the high concentration – wash and test with one or two other concentrations.

2. The demonstration of homologous interaction with the use of the two receptors, MOR and DOR, points directly at the receptors and not a downstream process that underlies the development of tolerance. This is a key control.

3. The construction of a straightforward method to monitor transmitter release using optical methods is a valuable advance for the study of all sorts of work on the molecular control of presynaptic release and the modulation of transmitter release.

*Reviewer #2 (Recommendations for the authors):*

The findings would be stronger if the authors could assess the expression level of transfected MOR and DOR and relate this to endogenous expression levels. This would address concerns about whether the observations are due to abnormally high or low levels of receptor expression in axons. This is important because depletion of a membrane reserve pool is posited to underlie slow desensitization. Thus overall expression level (assuming it correlates with membrane expression level) would be expected to contribute to the efficacy of agonist-driven membrane receptor depletion; likely the kinetics of the process and possibly the absolute degree of depletion as well.

Because some striatal MSNs express endogenous MOR and ~ half express endogenous DOR, one possibility is to repeat a few key experiments in untransfected neurons while restricting the analysis to presynaptic terminals identified to express endogenous opioid receptors (perhaps using a fluorescent ligand). Alternatively, can the authors make a valid claim about endogenous expression levels of MOR and DOR in individual striatal neurons and compare them to the levels achieved with transfection? If they differ substantially, it would be important to know how expression level impacts the desensitization demonstrated in Figure 1C.

*Reviewer #3 (Recommendations for the authors):*

1) Figure legend 1A(3) mentions that opioid receptors inhibit Ca^2+^ entry and subsequent exocytosis. In many presynaptic terminals, there is no evidence for opioid receptor inhibition of Ca^2+^ channels playing a role, in fact, it is often suggested that there is a direct role for βγ subunits. Is the change in fluorescence dependent on Ca^2+^? This technique may be a measure that can substantiate the role of Ca^2+^ and Ca^2+^ channels in opioid-mediated inhibition of neurotransmitter release. Alternatively, maybe remove this statement from the figure legend describing the diagram.

2) The authors should comment on whether the dynamic range and resolution of the fluorescence detection of endocytosis could capture endogenous opioid receptor signaling.

3) In the time course studies, p values are given on the right axis but appear to be generated from t-tests at each time point. Are these repeated measures on the same cultures with pictures taken at different agonists' times? If so, there should be a repeated ANOVA and the statistic for the group comparison is the only value necessary as it is clear which time points are statistically different. In other words, the graph is needlessly cluttered.

---

## [Author Response]

Essential revisions:1) Include discussions about the potential impact of overexpression of receptors and membrane protein dynamics.2) Incorporate the suggestions made by reviewers on the presentation.

We made the change suggested by the Editor, including a discussion on the impact of overexpression of receptors and changes on the presentation. We provide a new set of data suggested by the Reviewers, a concentration-response curve. We also provide a point by point response to the Reviewer’s critiques. We added an author to the list of contributors, Milos S. Simic, who significantly contributed to analysis of new data and statistical methods.

Reviewer #1 (Recommendations for the authors):1. The decline in the ability of DAMGO to reduce transmitter release following long-term incubation is clear, convincing, and well presented. It seems that in order to establish a strong conclusion for the development of tolerance the construction of even a rudimentary concentration-response curve following long-term agonist incubation would be valuable. That is long-term incubation with the high concentration – wash and test with one or two other concentrations.

This is a great point and something we overlooked. We performed the experiment suggested by reviewer 1, including concentration ranging from 0.1nM to 10uM. The resulting concentration-response curves show a reduction in potency and to a lesser degree in efficacy after chronic DAMGO treatment. We think this is consistent with our hypothesis that presynaptic tolerance is driven by depletion of the total axonal receptor reserve. We thank the reviewer for this suggestion and agree that it strengthens our mechanistic conclusions.

2. The demonstration of homologous interaction with the use of the two receptors, MOR and DOR, points directly at the receptors and not a downstream process that underlies the development of tolerance. This is a key control.3. The construction of a straightforward method to monitor transmitter release using optical methods is a valuable advance for the study of all sorts of work on the molecular control of presynaptic release and the modulation of transmitter release.

We appreciate that Reviewer 1 recognizes the rigor in our experimental approach and finds our methodology valuable to the community.

Reviewer #2 (Recommendations for the authors):The findings would be stronger if the authors could assess the expression level of transfected MOR and DOR and relate this to endogenous expression levels. This would address concerns about whether the observations are due to abnormally high or low levels of receptor expression in axons. This is important because depletion of a membrane reserve pool is posited to underlie slow desensitization. Thus overall expression level (assuming it correlates with membrane expression level) would be expected to contribute to the efficacy of agonist-driven membrane receptor depletion; likely the kinetics of the process and possibly the absolute degree of depletion as well.Because some striatal MSNs express endogenous MOR and ~ half express endogenous DOR, one possibility is to repeat a few key experiments in untransfected neurons while restricting the analysis to presynaptic terminals identified to express endogenous opioid receptors (perhaps using a fluorescent ligand). Alternatively, can the authors make a valid claim about endogenous expression levels of MOR and DOR in individual striatal neurons and compare them to the levels achieved with transfection? If they differ substantially, it would be important to know how expression level impacts the desensitization demonstrated in Figure 1C.

We agree that measurements of effects mediated by endogenous receptors would be preferable from the perspective of physiological relevance. We have indeed attempted to do this but, in striatal cultures prepared from the embryonic rat brain, we find that opioid receptors are endogenously expressed only in a small fraction of neurons – on the order of several percent. This is far from what is observed in the adult brain or acute slice preparations.

We deliberately use an electroporation method which results in recombinant expression levels lower than typically achieved using lipid-mediated transfection or viral transduction. In a previous study we assessed, using an anti-MOR antibody, the expression level of recombinant MOR achieved using our method in cultured striatal neurons to that of endogenous receptors in primary neurons prepared from several brain regions (striatum, hippocampus, habenula; Jullié et al., Neuron 2020, figure S1, cited in the manuscript). Our results suggested that electroporation yields levels similar to endogenous levels in cultured habenula neurons. Admittedly, this is a brain region in which a large number of neurons express MOR at a high level. Although we think this likely exceeds endogenous for some brain regions, we are confident that the expression levels are within the (wide) physiological range. We are also confident that our assay system can detect presynaptic inhibition mediated by endogenous MOR. We have verified this in habenula cultures at the overall population level (where this is possible due to a larger proportion of neurons expressing endogenous MOR). Based on our results with recombinant receptors, tolerance for MOR is associated with a ~30% reduction in the degree of MOR mediated presynaptic inhibition. The signal for presynaptic MOR inhibition, and its reduction after induction of tolerance, is therefore diluted to levels that make it experimentally challenging to distinguish from noise, even in habenula cultures. Accordingly, we view the present experimental model as one that is well suited for mechanistic investigation of presynaptic tolerance, but not one that necessarily replicates physiology.

We agree with the reviewer that the level of opioid receptor expression likely would impact the magnitude and/or kinetics of tolerance development. According to our model, and general pharmacological principles, we would expect overexpression to reduce the amount or rate of tolerance development. This is supported by our data indicating that reducing receptor reserve increases measured tolerance. Thus, if anything, overexpression of receptors would be expected to mask tolerance rather than accentuate it. Nevertheless, we detect profound tolerance, as is further supported by the new concentration-response data added to the revised manuscript. Thus we are confident that our experimental system models presynaptic tolerance and enables mechanistic investigation of this process, but we make no claim as to how the rate or magnitude of tolerance development measured in our cultured neuron system might relate to tolerance measured in a truly physiological system. In the revised manuscript we have attempted to be more clear about this important distinction, and we appreciate the Reviewer pointing it out.

Reviewer #3 (Recommendations for the authors):1) Figure legend 1A(3) mentions that opioid receptors inhibit Ca^2+^ entry and subsequent exocytosis. In many presynaptic terminals, there is no evidence for opioid receptor inhibition of Ca^2+^ channels playing a role, in fact, it is often suggested that there is a direct role for βγ subunits. Is the change in fluorescence dependent on Ca^2+^? This technique may be a measure that can substantiate the role of Ca^2+^ and Ca^2+^ channels in opioid-mediated inhibition of neurotransmitter release. Alternatively, maybe remove this statement from the figure legend describing the diagram.

In the current study, the change in fluorescence is thought to result from calcium dependent exocytosis of synaptic vesicles containing the VAMP2-SEP reporter. In a previous study (Jullié et al., Neuron 2020, Figure S4 D) we confirmed in a similar experimental system that extracellular calcium was required for this increase in fluorescence. As the Reviewer points out, using a reporter of Ca^2+^ instead of synaptic vesicle exocytosis would be one way to evaluate the contribution of Ca^2+^ entry blockade by opioid receptors on the inhibition of synaptic vesicle release. We note in this regard that a recent study did show opioid-mediated inhibition of calcium entry in terminals (He and al., *eLife* 2021, cited in the manuscript). Nevertheless, we agree that opioid receptors may inhibit synaptic transmission by more than one mechanism and that our data do not evaluate this. We thank the Reviewer for pointing this out, but note that our basic conclusions do not rely on a particular mechanism. Accordingly, in the revised manuscript, we have modified our diagram to simply remove the explicit depiction of calcium in an effort to avoid any over-interpretation.

2) The authors should comment on whether the dynamic range and resolution of the fluorescence detection of endocytosis could capture endogenous opioid receptor signaling.

The dynamic range of our system is indeed sufficient to detect presynaptic inhibition by endogenous receptors (Jullié et al., Neuron 2020 cited in the manuscript). However, as explained in response to Reviewer 2, the proportion of neurons that express endogenous opioid receptors in our embryonic striatal culture preparations is (surprisingly) low. This reduces the sensitivity of our system for endogenous receptor effects based on aggregate measurements. We added a paragraph in the revised version of the manuscript addressing these limitations.

3) In the time course studies, p values are given on the right axis but appear to be generated from t-tests at each time point. Are these repeated measures on the same cultures with pictures taken at different agonists' times? If so, there should be a repeated ANOVA and the statistic for the group comparison is the only value necessary as it is clear which time points are statistically different. In other words, the graph is needlessly cluttered.

Reviewer 3 is correct that the p-values were generated from independent t-tests at the different time points, and that time points represent repeated measures on the same cell. We also agree that the multiple statistics included needlessly cluttered the previous presentation. As the Reviewer suggested, we performed repeated measure ANOVA tests for all corresponding experiments. This analysis supports our conclusion for all but one condition, DOR + DADLE compared to DOR + vehicle, which is our strongest phenotype for agonist induced receptor internalization. The returned p-value is 0.0763, slightly above the significance threshold. We attribute this to one particular cell in the dataset (#3318) which exhibits an unusually high recruitment phenotype (>3 fold, ~2 folds the average measured recruitment). We recognize that the method employed for this quantification (based on calculation of fluorescence ratio) is subject to considerable cell to cell variability, in particular for cells that exhibit a very low baseline fluorescence over which quantification of the enrichment degree appears out of the normal range. This is the case upon review of primary measurements for this particular cell. The net effect is to add error despite the high enrichment value being fully consistent with our claim (agonist-induced accumulation of receptors in endosomes). For consistency, we decided to exclude from the statistical analysis all cells that exhibit such high (>3-fold) enrichment values. This results in exclusion of 3 cells in total and no more than any one condition. We explicitly explain this choice in the method section. The cells excluded by this criterion are DOR + agonist (#3318) compared to vehicle (p-value 0.0763 if included), MOR + agonist (#2782) compared to vehicle (p-value 0.0071 if included), DOR + DMSO (#3120) compared to DOR + Cmpd101 (p-value 0.48 if included). Accordingly, we believe that our exclusion is experimentally justified. It is also experimentally conservative in the case of the DOR + agonist condition (same trend as explained above) and, for the two other exclusions, there is no impact whatsoever.